# Sam68 promotes hepatic gluconeogenesis via CRTC2

Aijun Qiao [1], Junlan Zhou[2], Shiyue Xu[1], Wenxia Ma[1], Chan Boriboun[1], Teayoun Kim[3], Baolong Yan[1], Jianxin Deng[1], Liu Yang [1], Eric Zhang[1], Yuhua Song[1], Yongchao C. Ma [4], Stephane Richard [5], Chunxiang Zhang[1], Hongyu Qiu[6], Kirk M. Habegger[3], Jianyi Zhang [1] & Gangjian Qin [1,2 ✉]

Hepatic gluconeogenesis is essential for glucose homeostasis and also a therapeutic target for type 2 diabetes, but its mechanism is incompletely understood. Here, we report that Sam68, an RNA-binding adaptor protein and Src kinase substrate, is a novel regulator of hepatic gluconeogenesis. Both global and hepatic deletions of Sam68 significantly reduce blood glucose levels and the glucagon-induced expression of gluconeogenic genes. Protein, but not mRNA, levels of CRTC2, a crucial transcriptional regulator of gluconeogenesis, are >50% lower in Sam68-deficient hepatocytes than in wild-type hepatocytes. Sam68 interacts with CRTC2 and reduces CRTC2 ubiquitination. However, truncated mutants of Sam68 that lack the C- (Sam68$^{\Delta C}$) or N-terminal (Sam68$^{\Delta N}$) domains fails to bind CRTC2 or to stabilize CRTC2 protein, respectively, and transgenic Sam68$^{\Delta N}$ mice recapitulate the blood-glucose and gluconeogenesis profile of Sam68-deficient mice. Hepatic Sam68 expression is also upregulated in patients with diabetes and in two diabetic mouse models, while hepatocyte-specific Sam68 deficiencies alleviate diabetic hyperglycemia and improves insulin sensitivity in mice. Thus, our results identify a role for Sam68 in hepatic gluconeogenesis, and Sam68 may represent a therapeutic target for diabetes.

[1] Department of Biomedical Engineering, University of Alabama at Birmingham, School of Medicine and School of Engineering, Birmingham, AL, USA. [2] Feinberg Cardiovascular Research Institute, Northwestern University Feinberg School of Medicine, Chicago, IL, USA. [3] Department of Medicine - Endocrinology, Diabetes & Metabolism, University of Alabama at Birmingham, School of Medicine, Birmingham, AL, USA. [4] Departments of Pediatrics, Neurology and Physiology, Northwestern University Feinberg School of Medicine, Anne & Robert H. Lurie Children's Hospital of Chicago, Chicago, IL, USA. [5] Lady Davis Institute for Medical Research, McGill University, Montreal, QC, Canada. [6] Center for Molecular and Translational Medicine, Institute of Biomedical Science Georgia State University, Atlanta, GA, USA. ✉email: gqin@uab.edu

Of the estimated 34.3 million people in the US with diabetes, 90–95% have Type 2 (T2D), which is frequently accompanied by other comorbid diseases and contributes to the mortality of many chronic health conditions, including cardiovascular disease, stroke, and kidney disease[1]. In healthy individuals, blood glucose levels are stabilized via a balance between glucose consumption in the peripheral tissues and glucose production, ~90% of which occurs in the liver, and although the mechanisms of T2D are complex, hepatic glucose production is considered a first-line therapeutic target[2]. The liver produces glucose via two distinct processes, glycogenolysis and gluconeogenesis, and gluconeogenesis appears to be the major cause of elevated glucose production in T2D[3]; thus, drugs that downregulate gluconeogenesis, such as metformin, are among the most common treatments for hyperglycemia in patients with T2D[4–6].

Both glycogenolysis and gluconeogenesis are governed primarily by glucagon and insulin, which upregulate and downregulate glucose production, respectively[7,8]. The signaling mechanisms induced by these two counter-regulatory hormones converge on one member of a family of downstream transcriptional coactivators that are regulated by cyclic AMP response element (CRE) binding protein (CREB), and of the three known isoforms of CREB-regulated transcriptional coactivators (CRTC1, CRTC2, and CRTC3)[9], CRTC2 is most abundantly expressed in the liver[10]. In fasting animals, glucagon signaling increases and dephosphorylates CRTC2[11], which then translocates from the cytosol into the nucleus, where it binds the phosphorylated CREB and activates transcription of the master gluconeogenic regulator PPARγ coactivator 1α (PGC-1α) and the two enzymes that catalyze the rate-limiting steps of gluconeogenesis: phosphoenolpyruvate carboxykinase (PEPCK) and glucose-6-phosphatase (G6Pase)[12–14]. When feeding is re-initiated, the insulin pathway is upregulated, which promotes CRTC2 ubiquitination and degradation, thereby terminating gluconeogenesis[15]. Thus, since diabetic hyperglycemia occurs when glucagon levels (or the physiological sensitivity to glucagon) increase and/or insulin levels (or sensitivity) decline[16,17], strategies that manipulate the CREB/CRTC2 complex may be effective for the treatment of T2D.

Src-associated-in-mitosis-of-68kDa (Sam68; also known as KH-domain-containing, RNA-binding, signal-transduction-associated 1 [KHDRBS1]) is a member of the signal-transducer-and-activator-of-RNA (STAR) family of RNA-binding proteins[18,19] and participates in numerous cellular functions, including RNA processing[20,21], kinase- and growth-factor-signaling[22,23], transcription[24,25], cell-cycle regulation, and apoptosis[26,27]. Its range of activity is reflected in the wide variety of phenotypes observed in Sam68 knockout (Sam68$^{-/-}$) mice, which includes defects in spermatogenesis[28,29] and adipogenic differentiation[30], as well as a relatively lean body mass coupled with increases in adipose thermogenesis and an improved systemic glucose profile when fed a high-fat diet (HFD)[30,31]. Here, we show that Sam68 deletions, both globally and when restricted to the liver, reduce blood-glucose levels in mice by impeding gluconeogenesis, and that these effects are at least partially mediated by declines in CRTC2 protein stability and CRTC2/CREB-induced activation of gluconeogenic gene transcription. Our results also demonstrate that hepatic Sam68 deficiencies improve insulin sensitivity and reduce hyperglycemia in diabetic mice, which suggests that Sam68 could be a therapeutic target for the treatment of T2D.

## Results

### Sam68 regulates blood glucose homeostasis by promoting hepatic gluconeogenesis.

We initiated our investigation by comparing blood-glucose levels in Sam68$^{-/-}$ mice and their matched WT littermates; glucose levels were significantly lower in Sam68$^{-/-}$ mice under both feeding and fasting conditions (Fig. 1a), as well as in the pyruvate-tolerance (Fig. 1b) and glucagon-tolerance (Fig. 1c) tests (PTT and GcTT, respectively), both of which measure gluconeogenesis. Since gluconeogenesis occurs primarily in the liver, we generated hepatocyte-specific Sam68 knockout (Sam68$^{LKO}$) mice (Supplementary Fig. 1a–e), confirmed that Sam68 protein levels declined in the liver but not in other organs (Supplementary Fig. 1f), and then repeated our assessments: blood glucose levels were significantly lower in Sam68$^{LKO}$ mice than in their littermates with normal Sam68 expression (Sam68$^{f/f}$ mice), whether the animals were fed or fasted (Fig. 1d), and when evaluated in the PTT (Fig. 1e) and GcTT (Fig. 1f). Hepatic Sam68 deficiency was also associated with lower blood glucose levels after the injection of glucose (Fig. 1g) or insulin (Fig. 1h) (i.e., in the glucose-tolerance and insulin-tolerance tests [GTT and ITT, respectively]), and the amount of phosphorylated AKT (at serine-473 and tyrosine-308) in the liver was greater after the insulin treatment in Sam68$^{LKO}$ mice than in Sam68$^{f/f}$ mice (Fig. 1i). Notably, serum insulin concentrations were lower in Sam68$^{LKO}$ mice than in Sam68$^{fl/fl}$ littermates at both feeding and fasting conditions (Fig. 1j). Thus, hepatic Sam68 knockout appears to increase glucose tolerance and insulin sensitivity but reduce insulin levels.

The increase in insulin sensitivity observed in Sam68$^{LKO}$ mice was corroborated via hyperinsulinemic-euglycemic clamp assessments: when mice were systemically infused with insulin, the amount of glucose required to compensate for the increase in insulin levels and prevent hypoglycemia was significantly greater for Sam68$^{LKO}$ mice than for their Sam68$^{f/f}$ littermates (Fig. 1k). Furthermore, since plasma insulin levels in Sam68$^{LKO}$ mice and Sam68$^{f/f}$ mice were similar during clamping, the higher rate of glucose disposal observed in Sam68$^{LKO}$ mice (Fig. 1l) was attributable to differences in insulin sensitivity. Consistent with this whole-body increase in insulin sensitivity, the rate of glucose production was significantly lower in Sam68$^{LKO}$ mice than in Sam68$^{f/f}$ mice under clamped conditions (Fig. 1m) and was accompanied by a corresponding increase in hepatic glycogen synthesis (Fig. 1n). Thus, the hepatic-specific deletion of Sam68 appeared to reduce blood glucose levels by both increasing insulin sensitivity and reducing hepatic glucose production. Notably, serum triglyceride and free fatty acid levels in Sam68$^{f/f}$ and Sam68$^{LKO}$, fed or fasted, were similar (Supplementary Fig. 1g, h).

Sam68$^{-/-}$ mice are leaner than WT mice[31] and display defects in both adipocyte differentiation[30] and male fertility[32]. Thus, to verify that the effect of hepatic Sam68 deficiency on insulin sensitivity and gluconeogenesis was not caused by abnormalities in embryonic or neonatal development that subsequently altered liver function, assessments were conducted after inducing the Sam68 deletion in the hepatocytes of adult Sam68$^{f/f}$ mice via intravenous injection of adeno-associated virus 8 (AAV8) coding for the expression of Cre recombinase (AAV8-TBG-iCre) from the hepatocyte-specific thyroxine-binding globulin (TBG) promoter[33] (Sam68$^{f/f}$; AAV-Cre mice); control assessments were conducted in mice after administration of AAV8 coding for TBG-regulated GFP expression (Sam68$^{f/f}$; AAV-GFP mice) (Supplementary Fig. 1i). Three weeks after virus injections, blood-glucose levels were significantly lower in Sam68$^{f/f}$; AAV-Cre mice than in Sam68$^{f/f}$; AAV-GFP mice under both feeding and fasting conditions (Fig. 1o), and in the PTT (Fig. 1p), GcTT (Fig. 1q), GTT (Fig. 1r), and ITT (Supplementary Fig. 1j). Thus, the results from experiments with Sam68$^{f/f}$; AAV-Cre mice and Sam68$^{LKO}$ mice were consistent and demonstrate the role of hepatic Sam68 expression in systemic glucose homeostasis.

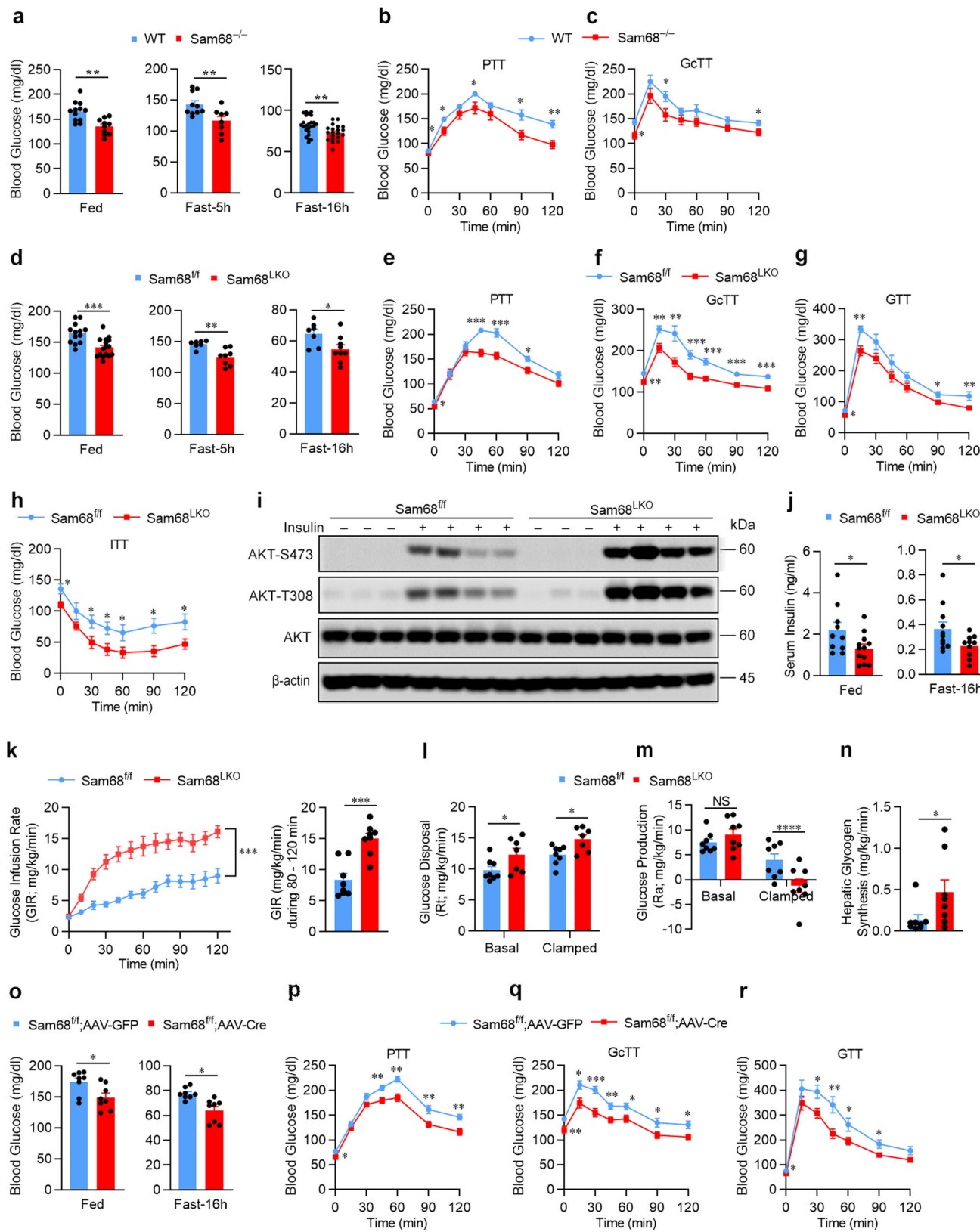

**Sam68 promotes hepatic gluconeogenesis by altering glucagon and insulin signaling balance.** To investigate the molecular mechanism by which Sam68 promotes hepatic gluconeogenesis, we analyzed mouse liver tissues for the expression of key gluconeogenic genes. Both mRNA (Fig. 2a) and protein (Fig. 2b) levels of PGC-1α, PEPCK, and G6Pase were dramatically lower in

Sam68LKO mice than in Sam68f/f mice under feeding and fasting conditions, while their FOXO1 and HNF4α protein levels were similar (Fig. 2b). Furthermore, when primary hepatocytes were isolated from Sam68−/− and WT mice and then treated with glucagon, the Sam68 deletion was associated with significantly lower measures of glucose production (Fig. 2c), and the

**Fig. 1 Hepatic Sam68 deficiency reduces blood glucose levels.** Blood glucose levels were measured in $Sam68^{-/-}$ and WT mice (**a**) under feeding conditions (WT, $n = 12$; $Sam68^{-/-}$, $n = 10$) and after the animals were fasted for 5 h (WT, $n = 10$; $Sam68^{-/-}$, $n = 8$) or 16 h (WT, $n = 24$; $Sam68^{-/-}$, $n = 20$), and at the indicated time points after the administration of (**b**) sodium pyruvate (pyruvate tolerance test [PTT]; WT, $n = 11$; $Sam68^{-/-}$, $n = 9$) or (**c**) glucagon (glucagon tolerance test [GcTT]; WT, $n = 10$; $Sam68^{-/-}$, $n = 9$). Blood glucose levels were measured in $Sam68^{LKO}$ and $Sam68^{f/f}$ mice (**d**) under feeding conditions ($Sam68^{f/f}$, $n = 13$; $Sam68^{LKO}$, $n = 16$) and after the animals were fasted for 5 h ($Sam68^{f/f}$, $n = 7$; $Sam68^{LKO}$, $n = 8$) or 16 h ($Sam68^{f/f}$, $n = 7$; $Sam68^{LKO}$, $n = 9$), (**e**) in the PTT ($Sam68^{f/f}$, $n = 7$; $Sam68^{LKO}$, $n = 9$) and (**f**) GcTT ($Sam68^{f/f}$, $n = 7$; $Sam68^{LKO}$, $n = 8$), and at the indicated time points after the administration of (**g**) glucose (glucose tolerance test [GTT]; $Sam68^{f/f}$, $n = 6$; $Sam68^{LKO}$, $n = 8$) or (**h**) insulin (insulin-tolerance test [ITT]; $Sam68^{f/f}$, $n = 7$; $Sam68^{LKO}$, $n = 8$). **i** Protein levels of phosphorylated AKT (at amino acids S473 and T308) and total AKT in the liver of $Sam68^{LKO}$ and $Sam68^{f/f}$ mice were evaluated via Western blot in 20 min after intraperitoneal injection of insulin (1 U/kg) (+) or saline (−). **j** Serum insulin levels in $Sam68^{LKO}$ and $Sam68^{f/f}$ mice at feeding condition ($Sam68^{f/f}$, $n = 10$; $Sam68^{LKO}$, $n = 12$) or after 16 h fasting ($Sam68^{f/f}$, $n = 11$; $Sam68^{LKO}$, $n = 10$). Hyperinsulinemic-euglycemic clamping studies were conducted in $Sam68^{LKO}$ and $Sam68^{f/f}$ mice. Mice were infused with radiolabeled ($[3-^3H]$) glucose for 120 min; then, insulin infusion was initiated and maintained at a constant rate for 120 min, and (**k**, left) the rate of $[3-^3H]$ glucose infusion was adjusted to maintain euglycemia (100 mg/dL). **k** (right) The glucose infusion rate (GIR) was calculated for the last 40 min of insulin infusion ($Sam68^{f/f}$, $n = 8$; $Sam68^{LKO}$, $n = 8$). Basal and clamped rates of glucose disposal (**l**; $Sam68^{f/f}$, $n = 8$; $Sam68^{LKO}$, $n = 7$) and glucose production (**m**; $Sam68^{f/f}$, $n = 8$; $Sam68^{LKO}$, $n = 8$) were calculated from the GIR and measurements of the plasma glucose specific activity. **n** Glycogen synthesis rate was determined by dividing the hepatic tracer glucose infusion rate (hepatic glycogen $^3H$ dpm/g liver) with plasma tracer glucose specific activity ($Sam68^{f/f}$, $n = 8$; $Sam68^{LKO}$, $n = 8$). $Sam68^{f/f}$ mice were injected with AAV8-TBG-iCre ($Sam68^{f/f}$; AAV-Cre, $n = 8$) to induce a hepatic-specific Sam68 deletion or with control AAV8-TBG-eGFP ($Sam68^{f/f}$; AAV-GFP, $n = 8$). Three weeks later, blood glucose levels were measured under (**o**) feeding conditions or after the animals were fasted for 16 h, and in the (**p**) PTT, (**q**) GcTT, and (**r**) GTT. Data are expressed as mean ± standard error of the mean (s.e.m.). *$p < 0.05$, **$p < 0.01$, ***$p < 0.001$, ****$p < 0.0001$ at the same time point, NS not significant (**a, d, j, n**, and **o**, and right panel of **k**: unpaired two-tailed $t$-test; **b, c, e–h, l, m, p–r**, and left panel of **k**: two-way ANOVA). Source data are provided as a Source Data file.

expression of all three gluconeogenic genes was significantly less upregulated (Fig. 2d, Supplementary Fig. 2a) in $Sam68^{-/-}$ hepatocytes than in WT hepatocytes, with the difference between the two cell populations, generally increasing in a time- and glucagon-dose-dependent manner. Notably, Sam68 is an RNA-binding protein, which suggests that it could influence gluconeogenic mRNA and protein levels by modulating mRNA stability; however, when WT and $Sam68^{-/-}$ hepatocytes were sequentially treated with glucagon and Actinomycin D (a specific RNA synthesis inhibitor), the stability of gluconeogenic mRNAs in the two cell populations was similar (Supplementary Fig. 2b). Thus, the Sam68 deletion appears to reduce gluconeogenic mRNA and protein levels of PGC-1α, PEPCK, and G6Pase by impeding glucagon signaling, rather than increasing the rate of their mRNA degradation.

**Sam68 potentiates glucagon signaling by maintaining CRTC2 protein levels.** The early steps in glucagon signaling are initiated by the binding of glucagon to the glucagon receptor, which promotes adenylyl cyclase activity; then, activated adenylyl cyclase increases cAMP production and cAMP-mediated protein kinase A (PKA) activation[34]. Sam68 deficiency did not significantly alter glucagon receptor expression (Fig. 2e–g) or mRNA levels for any of the subunits[35] of the PKA holoenzyme (Supplementary Fig. 2c, Fig. 2h) in mouse primary hepatocytes or in the livers of mice. Likewise, PKA activity, as well as the phosphorylation of PKA substrates, was similar in WT and $Sam68^{-/-}$ hepatocytes after the treatment with glucagon, forskolin (an adenylyl cyclase agonist), or Bt2-cAMP (a cAMP analog) (Fig. 2i, j, Supplementary Fig. 2d, e). Nevertheless, both glucose production and the expression of gluconeogenic genes were significantly less upregulated in $Sam68^{-/-}$ hepatocytes than in WT hepatocytes after the treatment with forskolin (Supplementary Fig. 2f, g) or Bt2-cAMP (Supplementary Fig. 2h, i). Thus, the upstream portion of the glucagon signaling pathway (i.e., from receptor binding through PKA activation) appears to be functionally isolated from Sam68.

The downstream components of glucagon signaling include CREB and CRTC2, which are phosphorylated and dephosphorylated, respectively, in response to PKA activation; then, dephosphorylated CRTC2 translocates into the nucleus, where it enhances the transcriptional activity of phospho-CREB and

gluconeogenic gene expression[14]. The Sam68 deletion did not substantially change phosphorylated or dephosphorylated CREB protein levels in hepatocytes, but CRTC2 protein levels (for all phosphorylation states) were lower in $Sam68^{-/-}$ than in WT hepatocytes after the treatment with glucagon (Fig. 3a), forskolin (Supplementary Fig. 3a), or Bt2-cAMP (Supplementary Fig. 3b). The amount of CRTC2 protein was also lower in the cytosolic and nuclear fractions of $Sam68^{-/-}$ hepatocytes than in the corresponding fractions of WT hepatocytes at baseline and after the glucagon treatment (Supplementary Fig. 3c). Chromatin immunoprecipitation (ChIP) assays confirmed that the lower CRTC2 protein levels observed in $Sam68^{-/-}$ hepatocytes were accompanied by dramatic declines in CRTC2 occupancy at the promoters of PGC-1α, G6Pase, and PEPCK after treatment with glucagon, forskolin, or Bt2-cAMP (Fig. 3b). However, the Sam68 deletion did not alter mRNA levels for any of the three CRTC isoforms (CRTC1, CRTC2, and CRTC3) in hepatocytes (Supplementary Fig. 3d) or in mouse liver tissues under feeding, fasting, or refeeding conditions (Supplementary Fig. 3e), and measures of CRTC1, CRTC2, and CRTC3 mRNA stability in $Sam68^{-/-}$ and WT hepatocytes were similar (Supplementary Fig. 3f). Furthermore, the protein levels of CRTC1 and CRTC3 in the liver of $Sam68^{LKO}$ mice and $Sam68^{f/f}$ littermates were similar under feeding or fasting condition (Supplementary Fig. 3g).

To confirm that the observed declines in CRTC2 protein levels contributed to the downregulation of glucagon signaling and gluconeogenesis in Sam68-deficient mice, experiments were conducted in $Sam68^{LKO}$ mice that had been injected with an adenovirus coding for a degradation-resistant variant of CRTC2 (Ad-CRTC2$^{K628R}$), which contained a Lys628Arg mutation at its major ubiquitination site[36] and has been shown to upregulate glucagon- and cAMP-agonist-induced gluconeogenic gene expression and glucose production in WT hepatocytes[15,36]. The vector was administered to $Sam68^{LKO}$ mice via tail-vein injection (i.e., in the $Sam68^{LKO}$; Ad-CRTC2$^{K628R}$ group), and comparative assessments were conducted in $Sam68^{LKO}$ mice treated with a GFP-encoding adenovirus (Ad-GFP) and in $Sam68^{f/f}$ mice treated with Ad-CRTC2$^{K628R}$ or Ad-GFP (i.e., the $Sam68^{LKO}$; Ad-GFP, $Sam68^{f/f}$; Ad-CRTC2$^{K628R}$, and $Sam68^{f/f}$; Ad-GFP groups, respectively) (Supplementary Fig. 3h). Under feeding (Fig. 3c) and fasting conditions (Fig. 3d), as well as in the PTT, GcTT, and GTT (Fig. 3e–g), blood glucose measurements in $Sam68^{LKO}$; Ad-CRTC2$^{K628R}$ mice were

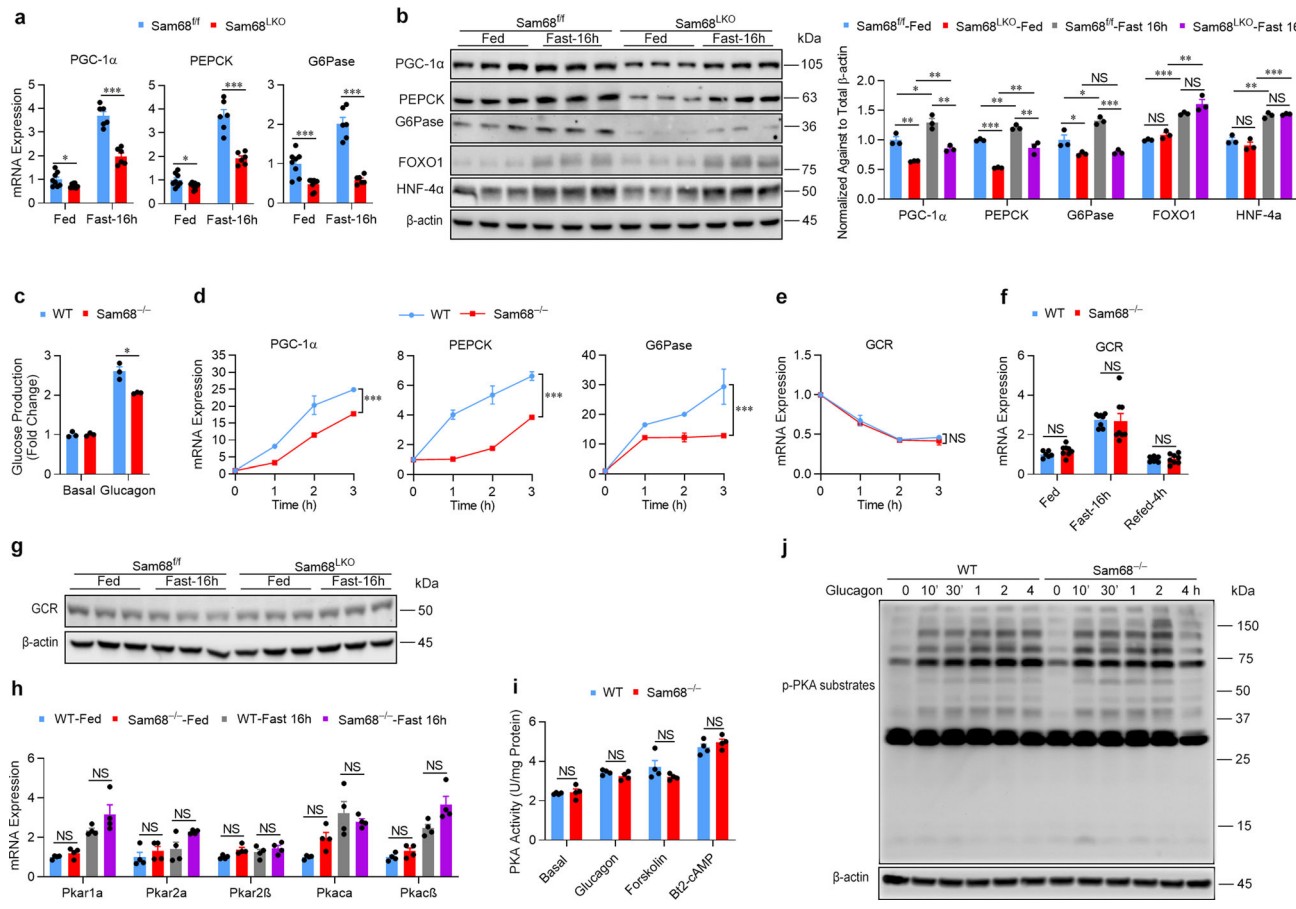

**Fig. 2 Hepatic Sam68 deficiency reduces glucagon signaling and gluconeogenic gene expression. a** mRNA and **b** protein expression of gluconeogenic genes were evaluated via qRT-PCR and Western blotting, respectively, in the livers of Sam68[LKO] and Sam68[f/f] mice under feeding conditions and after the animals had been fasted for 16 h (**a**; Fed, $n = 8$; Fast-16 h, $n = 6$. **b**; $n = 3$). **c** Glucose production was measured in WT and Sam68[−/−] primary hepatocytes that had been cultured with (Glucagon) or without (Basal) glucagon (100 nM) for 4 h ($n = 3$). **d** mRNA expression of gluconeogenic genes was measured in WT and Sam68[−/−] primary hepatocytes after treatment with glucagon for 0−3 h ($n = 3$). mRNA expression of the glucagon receptor (GCR) was evaluated (**e**) in WT and Sam68[−/−] primary hepatocytes after treatment with glucagon for 0−3 h ($n = 3$) and (**f**) in the livers of WT and Sam68[−/−] mice under feeding conditions (WT, $n = 6$; Sam68[−/−], $n = 8$), after fasting the animals for 16 h (WT, $n = 8$; Sam68[−/−], $n = 8$), and after 16 h of fasting followed by 4 h of refeeding (WT, $n = 8$; Sam68[−/−], $n = 8$). **g** Protein expression of GCR was assessed by Western blotting ($n = 3$). **h** mRNA expression of PKA subunits was evaluated in the livers of WT and Sam68[−/−] mice under feeding conditions or after 16 h of fasting ($n = 4$). **i** PKA activity was measured in WT and Sam68[−/−] primary hepatocytes after treatment with PBS (basal), glucagon, forskolin (10 μM), or Bt2-cAMP (100 μM) for 30 min ($n = 4$). **j** WT and Sam68[−/−] primary hepatocytes were treated with glucagon for 0−30 min and for 1−4 h; then, protein levels of phosphorylated PKA substrates were evaluated. Data are expressed as mean ± s.e.m. *$p < 0.05$, **$p < 0.01$, ***$p < 0.001$, NS, not significant (two-way ANOVA). "$n$" denotes biologically independent primary hepatocyte samples or liver tissues. Source data are provided as a Source Data file.

significantly greater than those in Sam68[LKO]; Ad-GFP mice and did not differ significantly from measurements in Sam68[f/f]; Ad-CRTC2[K628R] or Sam68[f/f]; Ad-GFP mice. CRTC2[K628R] expression also reversed the declines in mRNA (Fig. 3h) and protein (Fig. 3i) levels of gluconeogenic genes, as well as the increase in insulin-induced hepatic AKT phosphorylation (Supplementary Fig. 3i), observed in Sam68[LKO] mice, which is consistent with the role of CRTC2 in insulin sensitivity[37–39], and when CRTC2[K628R] was expressed in Sam68[−/−] hepatocytes (Supplementary Fig. 3j), measures of glucagon-, forskolin-, and Bt2-cAMP-induced glucose production (Supplementary Fig. 3k) and gluconeogenic gene expression (Supplementary Fig. 3l–n) increased significantly. Collectively, these observations indicate that the Sam68 deletion impedes glucagon signaling, reduces gluconeogenesis, and improves insulin sensitivity by reducing CRTC2 protein (but not mRNA) levels and the CRTC2-mediated activation of gluconeogenic gene expression. Consistently, overexpression of Sam68 markedly increased glucose production and gluconeogenic gene expression in WT but not CRTC2-knockdown HepG2 cells (Supplemental Fig. 3o–q), further

support that Sam68 promotes hepatic gluconeogenesis via CRTC2.

**Sam68 interacts with CRTC2 and suppresses CRTC2 degradation.** Because glucagon signaling induces gluconeogenesis, in part, by promoting the nuclear translocation of CRTC2[14], we investigated whether glucagon also altered the subcellular distribution of Sam68 in the livers of WT mice (Fig. 4a) and in cultured WT hepatocytes (Supplementary Fig. 4a): the total amount Sam68 protein was unaltered, but nuclear levels markedly increased, in response to glucagon treatment. Furthermore, Sam68 mediates a number of biological processes by functioning as an adaptor protein that interacts with other signaling molecules[40], and co-immunoprecipitation (co-IP) experiments confirmed that endogenous CRTC2 interacted with Sam68 in WT mouse primary hepatocytes at baseline (Fig. 4b) or with glucagon treatment (Supplementary Fig. 4b), and that exogenous Sam68 and CRTC2 interacted in 293 T cells that had been co-transfected with plasmids coding for hemagglutinin (HA)-tagged Sam68

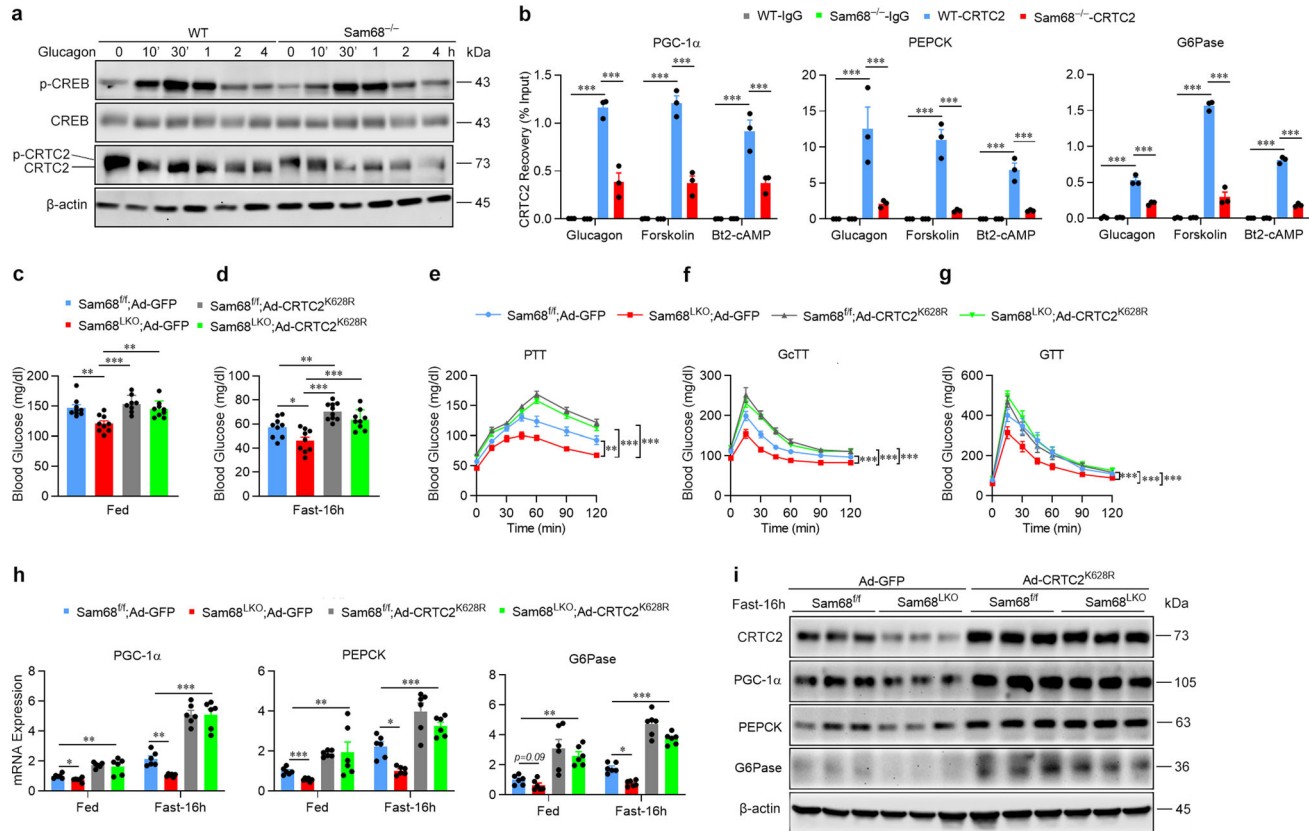

**Fig. 3 Sam68 enhances glucagon signaling by inhibiting CRTC2 degradation. a** WT and Sam68$^{-/-}$ primary hepatocytes were treated with glucagon (100 nM) for 0−30 min and 1−4 h; then, protein levels of phosphorylated CREB (p-CREB), total CREB, and CRTC2 were evaluated via Western blot. **b** WT and Sam68$^{-/-}$ hepatocytes were treated with glucagon, forskolin (10 μM), or Bt2-cAMP (100 μM) for 30 min; then, CRTC2 occupancy of the promoters for PGC-1α, PEPCK, and G6Pase was evaluated via chromatin immunoprecipitation (ChIP) assay ($n = 3$). **c–i** Sam68$^{LKO}$ and Sam68$^{f/f}$ mice were administered adenoviral vectors coding for GFP or a ubiquitination-defective CRTC2$^{K628R}$ mutant ($2 \times 10^9$ IFU per mouse). Four days later, blood glucose levels were measured (**c**) under feeding conditions (Sam68$^{f/f}$; Ad-GFP, $n = 9$. Sam68$^{LKO}$; Ad-GFP, $n = 9$. Sam68$^{f/f}$; Ad-CRTC2$^{K628R}$, $n = 8$; Sam68$^{LKO}$; Ad-CRTC2$^{K628R}$, $n = 9$), (**d**) after fasting for 16 h (Sam68$^{f/f}$; Ad-GFP, $n = 9$. Sam68$^{LKO}$; Ad-GFP, $n = 9$. Sam68$^{f/f}$; Ad-CRTC2$^{K628R}$, $n = 10$; Sam68$^{LKO}$; Ad-CRTC2$^{K628R}$, $n = 9$), in the (**e**) pyruvate-tolerance test (PTT; Sam68$^{f/f}$; Ad-GFP, $n = 9$. Sam68$^{LKO}$; Ad-GFP, $n = 9$. Sam68$^{f/f}$; Ad-CRTC2$^{K628R}$, $n = 10$; Sam68$^{LKO}$; Ad-CRTC2$^{K628R}$, $n = 11$), (**f**) glucagon-tolerance test (GcTT; Sam68$^{f/f}$; Ad-GFP, $n = 9$. Sam68$^{LKO}$; Ad-GFP, $n = 10$. Sam68$^{f/f}$; Ad-CRTC2$^{K628R}$, $n = 9$. Sam68$^{LKO}$; Ad-CRTC2$^{K628R}$, $n = 9$), and (**g**) glucose-tolerance test (GTT; Sam68$^{f/f}$; Ad-GFP, $n = 9$. Sam68$^{LKO}$; Ad-GFP, $n = 10$. Sam68$^{f/f}$; Ad-CRTC2$^{K628R}$, $n = 9$. Sam68$^{LKO}$; Ad-CRTC2$^{K628R}$, $n = 9$); and (**h**) mRNA and (**i**) protein levels of gluconeogenic genes in the liver were analyzed under feeding condition or after fasting for 16 h (**h**; $n = 6$. **i**; $n = 3$). Data are expressed as mean ± s.e.m.; **\****p < 0.05*, **\*\****p < 0.01*, **\*\*\****p < 0.001* (two-way ANOVA). "$n$" denotes biologically independent samples of primary hepatocytes, peripheral blood, or liver tissues. Source data are provided as a Source Data file.

(HA-Sam68) and FLAG-tagged CRTC2 (Flag-CRTC2) (Fig. 4c, Supplementary Fig. 4c). Notably, reverse co-IP experiments confirmed Sam68-CRTC2 interaction in mouse primary hepatocytes and further revealed that the interaction occurs in both the nucleus and cytoplasm (Supplementary Fig. 4d). We also identified which specific domains of Sam68 interact with CRTC2 by conducting co-IP experiments in 293T cells that had been co-transfected with Flag-CRTC2 and with plasmids coding for HA-tagged Sam68 truncation mutants lacking the N-terminal domain (amino acids 1–157, Sam68$^{\Delta N}$), the CK domain (amino acids 257–279, Sam68$^{\Delta CK}$), proline motifs 3 and 4 (amino acids 280–346, Sam68$^{\Delta P3\text{-}P4}$), or the C-terminal domain (amino acids 347–443, Sam68$^{\Delta C}$); Flag-CRTC2 failed to bind HA-Sam68$^{\Delta C}$ and interacted much less strongly with HA-Sam68$^{\Delta N}$ and HA-Sam68$^{\Delta CK}$ than with full-length HA-Sam68 (Fig. 4d, e).

Our observation that the amount of CRTC2 protein, but not mRNA, was significantly downregulated in Sam68$^{-/-}$ hepatocytes suggests that the interaction between Sam68 and CRTC2 reduces the rate of CRTC2 degradation. We tested this hypothesis by measuring CRTC2 protein levels in WT and Sam68$^{-/-}$ hepatocytes after the cells had been treated with cycloheximide

for 0–8 h to inhibit new protein synthesis: CRTC2 protein levels declined more rapidly in Sam68$^{-/-}$ hepatocytes (Fig. 4f). In contrast, overexpression of Sam68 in WT hepatocytes with Adv-Sam68 slowed the decline of CRTC2 proteins (Supplemental Fig. 4e), which indicates that Sam68 promotes CRTC2 protein stability. Furthermore, since protein degradation occurs primarily through the ubiquitin-proteasome system or the autophagy-lysosome pathway[41], we also monitored CRTC2 protein levels in WT and Sam68$^{-/-}$ hepatocytes after up to 16 h of treatment with each of the two proteasome inhibitors, MG132 or BZM, and up to 8 h of treatment with the autophagy inhibitor bafilomycin. CRTC2 protein levels approximately doubled in Sam68$^{-/-}$ hepatocytes during the MG132- and BZM-treatment periods (Fig. 4g), but were unchanged by bafilomycin (Fig. 4h), and none of the three treatments altered CRTC2 protein levels in WT hepatocytes by more than ~10%. These results were corroborated by immunoprecipitating CRTC2 from WT and Sam68$^{-/-}$ hepatocytes after 16 h of treatment with MG132 (when the total amount of CRTC2 protein in the two cell populations was equivalent), and then measuring the amount of ubiquitinated protein in the precipitate: polyubiquitinated CRTC2 protein levels

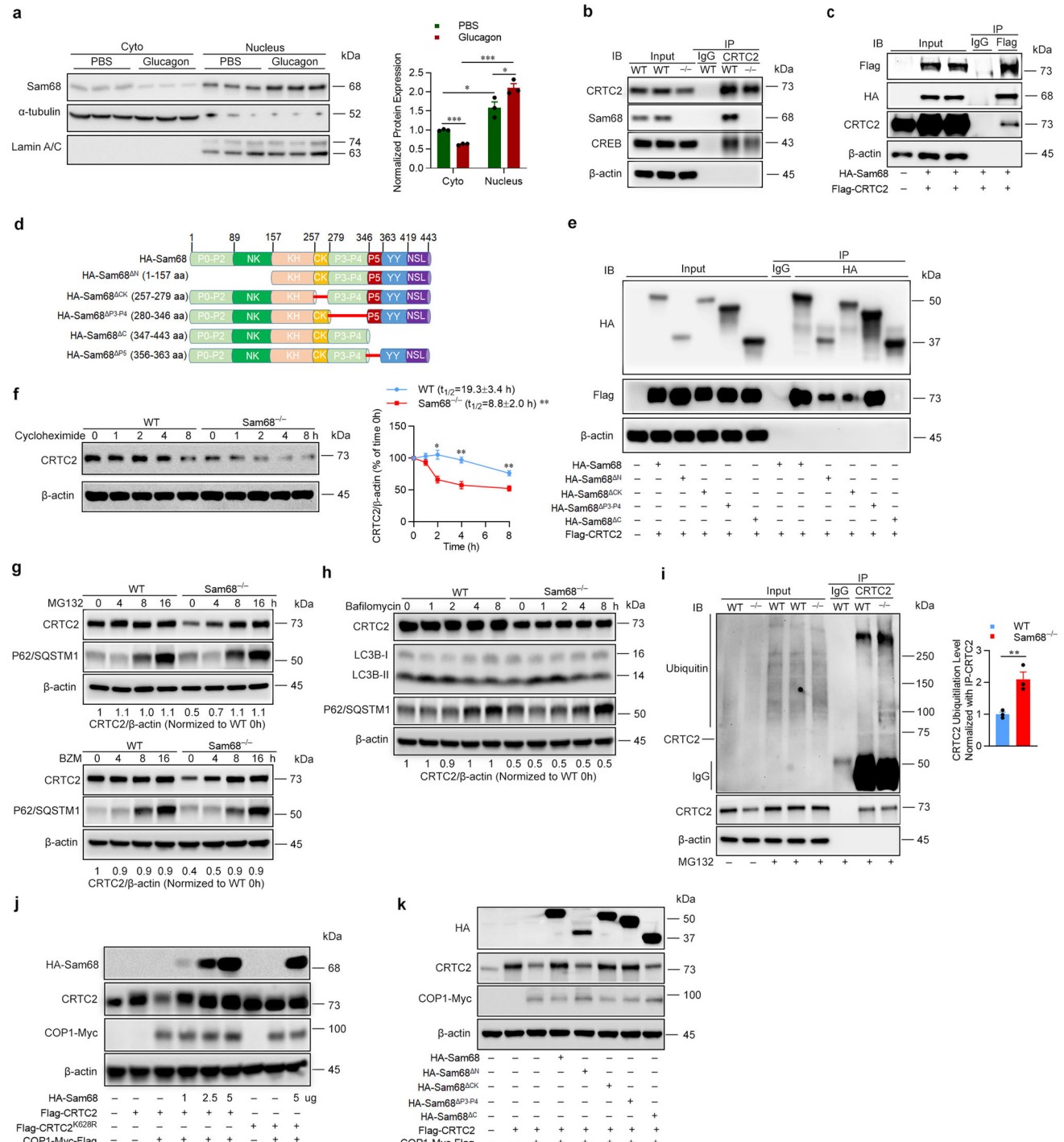

were greater in Sam68$^{-/-}$ cells than in WT cells (Fig. 4i). Thus, Sam68 appears to promote CRTC2 stability by impeding CRTC2 ubiquitination and proteasomal degradation.

The ubiquitination of CRTC2 is primarily mediated by the E3 ligase constitutive photomorphogenic protein 1 (COP1)[15,36], and when HepG2 cells were cotransfected with three plasmids—one coding for Myc-tagged COP1 (COP1-Myc), one for Flag-CRTC2, and one for HA-Sam68, with the HA-Sam68 plasmid delivered in progressively greater amounts—CRTC2 protein levels grew as the amount of HA-Sam68 protein increased (Fig. 4j); notably, CRTC2 protein levels remained largely stable in cells cotransfected with COP1-Myc, with a plasmid coding for a Flag-tagged version of the degradation-resistant CRTC2 variant (Flag-CRTC2$^{K628R}$),

and with or without HA-Sam68. To determine which domain of Sam68 mediates the decline in COP1-induced CRTC2 ubiquitination, CRTC2 protein levels were evaluated in HepG2 cells that had been cotransfected with COP1-Myc and Flag-CRTC2, and with HA-Sam68 or each of the HA-tagged Sam68 truncation mutants: both Sam68$^{\Delta N}$ and Sam68$^{\Delta C}$ mutations were associated with a decline in the amount of intact CRTC2 protein, however, similar levels of CRTC2 decline reached with a much lesser Sam68$^{\Delta N}$ than Sam68$^{\Delta C}$ (Fig. 4k and Supplementary Fig. 4f). Notably, Sam68$^{\Delta N}$ reduced CRTC2 levels in a dose-dependent manner (Supplementary Fig. 4f). Since the interaction between Sam68 and CRTC2 is primarily mediated by the C-terminus of Sam68 (Fig. 4e), which is preserved in the Sam68$^{\Delta N}$ mutant, the

**Fig. 4 Sam68 binds CRTC2 and inhibits CRTC2 ubiquitination and proteosomal degradation. a** WT mice were treated with glucagon (30 μg/kg) or PBS for 10 min; then, nuclear and cytoplasmic proteins were isolated from the liver tissue, and Sam68 protein expression was evaluated via Western blot ($n = 3$). **b** CRTC2 was immunoprecipitated from WT and $Sam68^{-/-}$ primary hepatocytes, and then CRTC2, Sam68, and CREB protein was detected in the precipitates via immunoblot. **c** 293T cells were co-transfected with plasmids coding for HA-tagged Sam68 and Flag-tagged CRTC2; 48 h later, CRTC2 was immunoprecipitated from the cells with anti-Flag, and Sam68 was detected in the precipitates by immunoblotting with anti-HA. **d** Schematic presentation of Sam68 domain structures and mutants used in this report. **e** 293T cells were co-transfected with plasmids coding for Flag-tagged CRTC2 and HA-tagged WT Sam68 or HA-tagged mutations of Sam68 lacking the N-terminal domain (ΔN), the CK domain (ΔCK), proline motifs 3 and 4 (ΔP3-P4), or the C-terminal domain (ΔC); 48 h later, Sam68 and the Sam68 mutants were immunoprecipitated with anti-HA, and CRTC2 was detected in the precipitates by immunoblotting with anti-Flag. **f** WT and $Sam68^{-/-}$ primary hepatocytes were treated with cycloheximide (100 μM) for 0−8 h to block new protein synthesis; then, CRTC2 protein levels were evaluated by Western blot, quantified via NIH Image J software, normalized to β-actin levels, and reported as the proportion of the amount present at 0 h ($n = 3$). The CRTC2 half-lives ($t_{1/2}$) were calculated using GraphPad Prism8 Software. WT and $Sam68^{-/-}$ primary hepatocytes were treated with (**g**) MG132 (20 μM) or BZM (100 nM) for 0−16 h, or (**h**) bafilomycin (100 nM) for 0−8 h; then, CRTC2 protein levels were evaluated by Western blot, quantified via Image J software, normalized to β-actin levels, and reported as the proportion of the amount present in WT cells at 0 h. P62/SQSTM1 and LC3BI/II protein levels were also evaluated to indicate autophagic activity. **i** WT and $Sam68^{-/-}$ primary hepatocytes were treated with MG132 (20 μM) for 16 h; then, CRTC2 was immunoprecipitated from the cells, and ubiquitinated proteins were detected in the immunoprecipitates via immunoblot (left, representative; right, quantification from three independent experiments). **j** HepG2 cells were co-transfected with plasmids coding for Flag-tagged CRTC2, Myc-Flag-tagged COP1, and HA-tagged WT Sam68; 48 h later, Sam68, CRTC2, and COP1 protein levels were evaluated via immunoblotting with anti-HA, anti-CRTC2, and anti-Myc antibodies, respectively. **k** HepG2 cells were co-transfected with three plasmids, one coding for Flag-tagged CRTC2, one for Myc-Flag-tagged COP1, and one for HA-tagged WT Sam68 or for each of the HA-tagged Sam68 truncation mutants; 48 h later, protein levels for CRTC2, COP1, and Sam68 or the Sam68 truncation mutants were evaluated via immunoblotting with anti-CRTC2, anti-Myc, and anti-HA antibodies, respectively. Data are expressed as mean ± s.e.m. *$p < 0.05$, **$p < 0.01$, ***$p < 0.001$ (right panels of **a** and **f**: two-way ANOVA; right panel of **i**: unpaired two-tailed *t*-test). "*n*" denotes biologically independent primary hepatocyte samples or liver tissues. Source data are provided as a Source Data file.

N-terminal truncation (ΔN) appears to be a dominant-negative mutation that competes with the WT Sam68 protein for CRTC2 binding but fails to impede COP1-induced ubiquitination. Furthermore, computational results from the combined text pattern search and hydropathy analyses showed that the P5 motif located near the C-terminus of Sam68 has 87.5% hydropathic complementarity and 0.461° of complementary hydropathy with N-terminal nuclear localization domain of CRTC2 (amino acids 77–84) in a palindromic manner (Supplementary Fig. 4g, h), suggesting that P5 motif in Sam68 likely binds amino acids 77–84 in CRTC2 (Supplementary Fig. 4i). Indeed, Sam68 with P5 truncation (HA-Sam68$^{ΔP5}$) (Fig. 4d) failed to interact with Flag-CRTC2 in co-IP experiments (Supplementary Fig. 4j), confirming that the P5 motif is essential for the Sam68−CRTC2 interaction.

**Sam68$^{ΔN}$ transgenic mice mimic the changes in glucose metabolism and insulin sensitivity observed in Sam68$^{-/-}$ mice.** Because the Sam68$^{ΔN}$ truncation disrupted Sam68-CRTC2 binding and reduced CRTC2 stability, we investigated whether the changes in blood-glucose levels, gluconeogenic gene expression, and CRTC2 protein levels observed in Sam68$^{-/-}$ and Sam68$^{LKO}$ mice were reproduced in a line of transgenic, Sam68$^{ΔN}$ mutant (Sam68$^{ΔN-Tg}$) mice (Supplementary Fig. 5a, b). Compared to their WT littermates, Sam68$^{ΔN-Tg}$ mice displayed lower blood-glucose levels under fed and fasting conditions (Fig. 5a) and in the PTT, GcTT, GTT, and ITT (Fig. 5b–d, Supplementary Fig. 5c); declines in fed and fasted mRNA (Fig. 5e) and protein (Fig. 5f) measurements of gluconeogenic gene expression; upregulated hepatic insulin signaling (Supplementary Fig. 5d); and lower amounts of intact CRTC2 protein (Fig. 5f). Thus, Sam68$^{ΔN-Tg}$ mice were phenotypically similar to Sam68$^{-/-}$ and Sam68$^{LKO}$ mice, which emphasizes the importance of the N-terminal domain of Sam68 in glucose metabolism, insulin sensitivity, and glucagon signaling.

**Downregulating hepatic Sam68/CRTC2 signaling mitigates hyperglycemia in diabetic mice.** Our observation that the Sam68 deletion lowers blood-glucose levels, and promotes insulin

sensitivity by decreasing the stability of CRTC2 protein, suggests that these two molecules may also have a role in the pathogenic mechanisms of diabetes. We began to test this hypothesis by determining whether hepatic Sam68 and/or CRTC2 were upregulated in two commonly used diabetic models, HFD-fed mice, and *db/db* mice, and in human subjects with or without diabetes. In both the mouse models, cytosolic and nuclear levels of Sam68 and CRTC2 protein (Fig. 6a, b), as well as mRNA measurements of CRTC2 and gluconeogenic gene expression (Supplementary Fig. 6a, b), were significantly greater in liver cells from diabetic than from non-diabetic animals, but only diabetic HFD-fed mouse hepatocytes displayed increases in *Sam68* mRNA. Elevated amounts of Sam68 and CRTC2 protein (Fig. 6c), but not mRNA (Supplementary Fig. 6c), were also observed in liver tissues from patients with diabetes and were accompanied by a significant increase in both mRNA and protein levels of gluconeogenic genes.

To confirm that the diabetic phenotypes of HFD-fed and *db/db* mice could be at least partially attributed to the observed increase in Sam68 expression, experiments were conducted in Sam68$^{LKO}$ and Sam68$^{fl/fl}$ that had been fed via the HFD protocol to induce diabetes, and in *db/db* mice after injection with AAV8 coding for Sam68 shRNA (*db/db*; sh-Sam68 mice) or a scrambled sequence (*db/db*; sh-Scr); the Sam68 shRNA reduced Sam68 mRNA and protein levels in the liver by 77% and 70%, respectively, but not in other organs (Fig. 6d, e, Supplementary Fig. 6d). Compared to their corresponding control groups, Sam68$^{LKO}$ and *db/db*; sh-Sam68 mice had significantly lower blood glucose levels under feeding and fasting conditions (Fig. 6f, Supplementary Fig. 6e) and in the PTT (Fig. 6g), GcTT (Fig. 6h), GTT (Fig. 6i), and ITT (Supplementary Fig. 6f); significantly lower mRNA (Fig. 6j) and protein (Fig. 6k) measurements of gluconeogenic gene expression; significantly reduced CRTC2 protein levels (Fig. 6k); and significant increases in insulin signaling (Supplementary Fig. 6g) and decreases in serum insulin levels (Supplementary Fig. 6h). Collectively, these observations demonstrate that the hyperglycemic phenotypes in HFD-fed and *db/db* mice can be alleviated by downregulating Sam68 expression in the liver and,

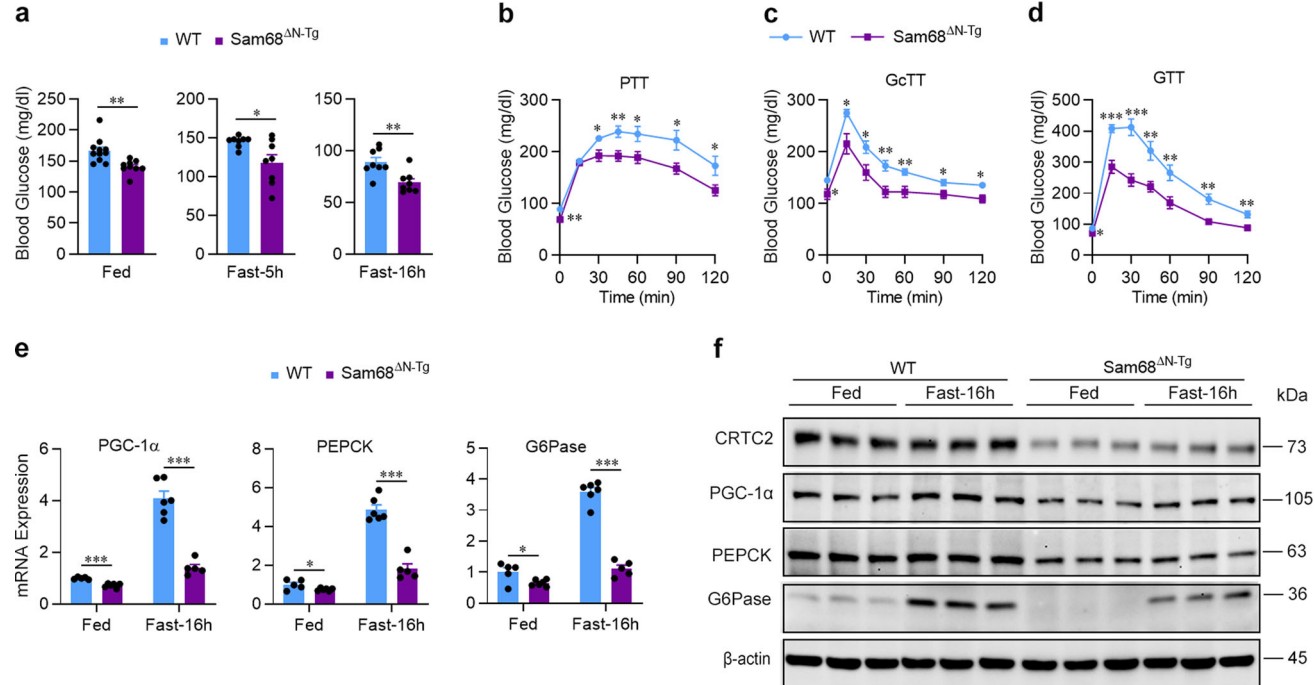

**Fig. 5 Blood glucose levels and gluconeogenic gene expression are lower in mice carrying an N-terminal deletion (ΔN) of Sam68 than in WT mice.** Blood glucose levels were measured in Sam68$^{ΔN-Tg}$ and their WT littermates (**a**) under feeding conditions (WT, $n = 14$; Sam68$^{ΔN-Tg}$, $n = 9$) and after fasting for 5 h or 16 h (WT, $n = 8$; Sam68$^{ΔN-Tg}$, $n = 8$), and in the (**b**) pyruvate tolerance test (PTT; WT, $n = 8$; Sam68$^{ΔN-Tg}$, $n = 8$), (**c**) glucagon tolerance test (GcTT; WT, $n = 8$; Sam68$^{ΔN-Tg}$, $n = 8$), and (**d**) glucose tolerance test (GTT; WT, $n = 9$; Sam68$^{ΔN-Tg}$, $n = 8$). Sam68$^{ΔN-Tg}$ and WT mice were sacrificed under feeding conditions or after fasting for 16 h; then, (**e**) mRNA and (**f**) protein expression of gluconeogenic genes, and protein levels of CRTC2, were evaluated in liver tissues (**e**; Fed: WT, $n = 5$; Sam68$^{ΔN-Tg}$, $n = 6$. Fast-16 h: WT, $n = 6$; Sam68$^{ΔN-Tg}$, $n = 5$. **f**; $n = 3$). Data are expressed as mean ± s.e.m. *$p < 0.05$, **$p < 0.01$, ***$p < 0.001$ versus WT mice under the same condition and at the same time point (**a**: unpaired two-tailed $t$-test; **b**–**e**: two-way ANOVA). Source data are provided as a Source Data file.

consequently, that Sam68 may be a therapeutic target for the treatment of T2D.

## Discussion

The elevations in blood glucose associated with T2D occur through a combination of declines in insulin sensitivity, which reduces peripheral glucose uptake, and an increase in hepatic glucose production. Many first-line therapies for T2D, target the increase in production, particularly via hepatic gluconeogenesis[2,42,43]. While an improved systemic glucose profile in HFD-fed Sam68$^{−/−}$ mice was previously reported by us[30,31], the results presented here are the first to show that Sam68 has a key regulatory role in gluconeogenesis and blood glucose maintenance. Global and liver-specific deletions of Sam68 in mice reduced blood glucose levels under feeding and fasting conditions, as well as in the PTT, GcTT, GTT, and ITT, and these declines were accompanied by a drop in CRTC2 protein levels and in the expression of key gluconeogenic regulators that are upregulated by CRTC2 in response to glucagon signaling. Furthermore, hepatic Sam68 expression was upregulated both in patients with diabetes and in two diabetic mouse models (HFD-fed and *db/db*), and the hyperglycemic phenotype observed in diabetic mice significantly improved when hepatic Sam68 expression was reduced via genetic deletion or shRNA-mediated inhibition. Thus, the therapies targeting the expression of Sam68 may be effective for normalizing blood-glucose levels in patients with diabetes.

The important role of hepatic CRTC2 in gluconeogenesis and insulin sensitivity has been extensively documented. In humans, CRTC2 polymorphisms are associated with an increased risk for T2D[44,45], and when CRTC2 levels were manipulated in mice, the

corresponding changes in blood-glucose measurements were consistent with our observations in Sam68-deficient mice: modest elevations of CRTC2 increased blood glucose levels and decreased insulin sensitivity[15,37], global or liver-specific CRTC2 deletions reduced blood glucose and increased insulin sensitivity[15,39,46], and both oligo- and siRNA-mediated CRTC2 inactivation mitigated hyperglycemia in diabetic mice[47,48]. CRTC2 triggers both glucose production and a compensatory mechanism that elevates insulin levels to promote glucose uptake[15,37,39], which supports the notion that the reduced serum insulin levels in Sam68$^{LKO}$ mice may be secondary to altered blood glucose. Additionally, CRTC2 has been shown to modulate insulin sensitivity by regulating lipid metabolism[38], and interestingly both the global Sam68 deletion[31] and the loss of hepatic CRTC2 expression activate thermogenesis[46]. Thus multiple mechanisms may contribute to the enhanced insulin sensitivity with hepatic Sam68-deficiency, which warrants further investigations.

Our data demonstrate that Sam68 interacts with CRTC2 and prevents the COP1-mediated polyubiquitination and proteasomal degradation of CRTC2 in hepatocytes, which is consistent with the observation that COP1 ubiquitinates CRTC2, and that this mechanism is closely associated with CRTC2-mediated gluconeogenic activity[15,36]. We also showed that both the C- and N-termini of Sam68 are required for CRTC2 stabilization: the C-terminal truncation disrupted the Sam68−CRTC2 interaction, and the ΔN mutant failed to increase CRTC2 stability. The role of the C-terminus in binding was also supported by our hydropathy modeling and P5 mutagenesis experiments. However, the N-terminus of Sam68 is not predicted with a high probability to interact with CRTC2, suggesting a more complex mechanism and potential involvement of other molecular components.

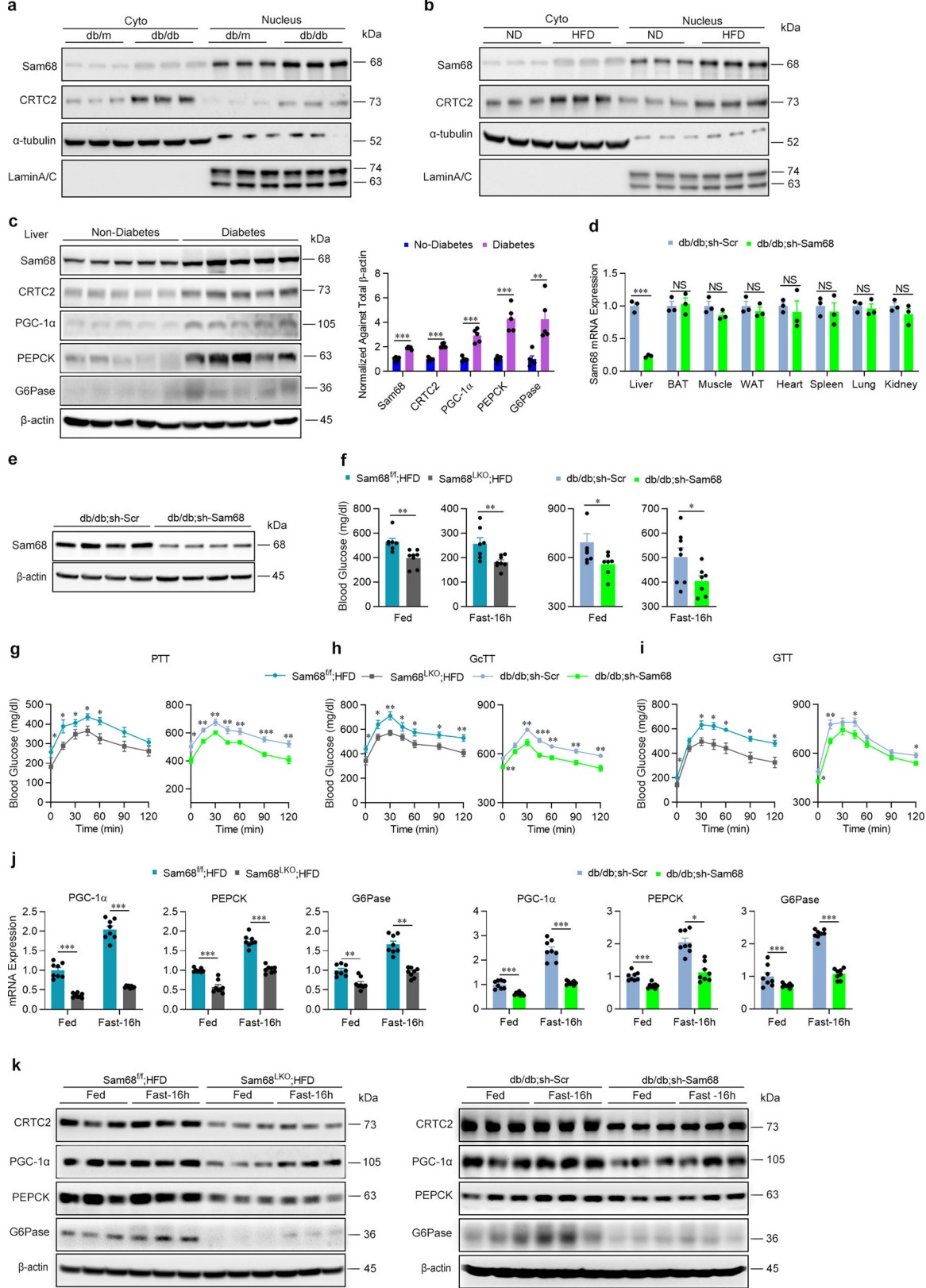

We observed that hepatic Sam68 protein expression is significantly upregulated in diabetic mouse models and in human subjects with diabetes, that glucagon signaling promotes Sam68 translocation from cytoplasm to nucleus, and that Sam68 interacts with CRTC2 both in the nucleus and cytoplasm. Evidences from other laboratories suggest that insulin induces nuclear export of Sam68 in rat adipocytes[49,50], and a variety of posttranscriptional modifications, including tyrosine phosphorylation[18], serine/ threonine phosphorylation[51], acetylation[52], methylation[53], and SUMOylation[54], are associated with Sam68 localization and stability. Thus, it is likely that under diabetic conditions, deregulated metabolic signaling (e.g., glucagon and insulin) result in skewed

**Fig. 6 Hepatic Sam68 inactivation mitigates diabetic hyperglycemia.** Sam68 and CRTC2 protein expression were evaluated in the nucleus and cytoplasm of liver cells (**a**) from WT mice fed a normal diet (ND) and diabetic WT mice fed an HFD for 3 months and (**b**) from *db/db* and control *db/m* mice at age of 2–3 months (*n* = 3). **c** Sam68, CRTC2, PGC-1α, PEPCK, and G6Pase protein levels in the livers of patients with or without diabetes were evaluated via Western blot (left), quantified densitometrically via NIH Image J software, normalized to *β*-actin levels, and expressed relative to the levels in non-diabetic patients (right). *n* = 5 per group. *db/db* (5-week-old) mice were administered AAV8 coding for GFP-labeled murine Sam68 shRNA (*db/db*; sh-Sam68 mice) or for GFP-labeled scrambled Sam68 shRNA (*db/db*; sh-Scr mice) (1 × 10$^{12}$ genome copies per mouse); 3 weeks later, Sam68 (**d**) mRNA and (**e**) protein levels were evaluated via qRT-PCR (liver, brown fat, skeletal muscle, epididymal fat, heart, spleen, lung, and kidney; *n* = 3) and Western blot (liver; *n* = 4), respectively. **f–i** Blood glucose levels were measured in diabetic, HFD-fed Sam68$^{f/f}$ (Sam68$^{f/f}$; HFD) and Sam68$^{LKO}$ (Sam68$^{LKO}$; HFD) mice and in *db/db*; sh-Scr and *db/db*; sh-Sam68 mice (**f**) under feeding conditions and after fasting for 16 h (Sam68$^{f/f}$; HFD: *n* = 7, Sam68$^{LKO}$; HFD: *n* = 8, *db/db*; sh-Scr: *n* = 8, *db/db*; sh-Sam68: *n* = 7), and in the (**g**) pyruvate-tolerance test (PTT; Sam68$^{f/f}$; HFD: *n* = 7, Sam68$^{LKO}$; HFD: *n* = 8, *db/db*; sh-Scr: *n* = 8, *db/db*; sh-Sam68: *n* = 8), (**h**) glucagon-tolerance test (GcTT; Sam68$^{f/f}$; HFD: *n* = 7, Sam68$^{LKO}$; HFD: *n* = 9, *db/db*; sh-Scr: *n* = 8, *db/db*; sh-Sam68: *n* = 8), and (**i**) glucose-tolerance test (GTT; Sam68$^{f/f}$; HFD: *n* = 7, Sam68$^{LKO}$; HFD: *n* = 9, *db/db*; sh-Scr: *n* = 8, *db/db*; sh-Sam68: *n* = 8). **j** mRNA and (**k**) protein levels of gluconeogenic genes were evaluated in the livers of Sam68$^{f/f}$; HFD and Sam68$^{LKO}$; HFD mice and of *db/db*; sh-Scr and *db/db*; sh-Sam68 mice (**j**; *n* = 8. **k**; *n* = 3). Data are expressed as mean ± s.e.m. *\*p* < 0.05, \*\**p* < 0.01, \*\*\**p* < 0.001, NS, not significant (**d**, **f**, right panel of **c**: unpaired two-tailed *t*-test; **g–j**: two-way ANOVA). "*n*" denotes biologically independent human liver samples, or mouse blood or tissue samples. Source data are provided as a Source Data file.

posttranscriptional modifications and subcellular trafficking of Sam68 to increase CRTC2 stability, and studies to identify these site-specific modifications may provide additional mechanistic insights and therapeutic targets of T2D.

As an adaptor protein, Sam68 coordinates various cellular responses to environmental stimuli. Notably, we and others have demonstrated its potent effects in mediating TNF-α receptor signaling and NF-κB activation in a number of cell types[23,24,55]. Since T2D are complex conditions where sub-acute and chronic inflammation plays a crucial role via the production of pro-inflammatory cytokines and activation of major inflammatory pathways, particularly TNF-α and NF-κB[56,57], our findings suggest that Sam68 may provide a mechanistic link between tissue inflammation and the pathophysiology of diabetic hyperglycemia.

In conclusion, the evidence presented in this report demonstrates that Sam68 promotes hepatic and gluconeogenesis by interacting with and stabilizing CRTC2 protein. We also show that the hyperglycemic phenotype observed in two mouse models of diabetes declined significantly in response to hepatic Sam68 inactivation, which suggests that novel therapies targeting Sam68 and/or the Sam68−CRTC2 interaction may normalize blood-glucose levels in patients with diabetes.

## Methods

**Animal studies.** All animal experiments in this report were approved by the Institutional Animal Care and Use Committees (IACUC) of Northwestern University and the University of Alabama at Birmingham and comply with all relevant ethical regulations, including the National Institutes of Health (NIH) "Guide for the Care and Use of Laboratory Animals". Experiments were conducted in 2- to 3-month-old male mice on C57BL/6J background unless specified otherwise. Mice were fed ad libitum and maintained in a climate-controlled facility (22 °C, 43% humidity) with a 12:12-h light:dark cycle.

**Genetic mice.** Sam68$^{−/−}$ mice were generated as previously described[58]; *db/db* and *db/m* mice were purchased from Jackson Laboratory (No. 000642).

The pGK-Sam68$^{floxEx5-8}$ targeting vector used to generate Sam68$^{flox/flox}$ mice was constructed from the pGKneoF2L2DTA plasmid; the neomycin gene and distal loxP site were inserted into intron 8, and the proximal loxP site was inserted into intron 4 of the Sam68 allele. Briefly, three fragments—a 5′ arm fragment (1736 bp) containing 5′ SacII and 3′ NotI restriction sites, a central fragment (4888 bp) consisting of the floxed allele of exons 5–8 flanked by loxP sites containing 5′ Asc I and 3-Fse I restriction sites, and a 3′ arm fragment (1771 bp) containing 5′ NheI and 3′ SalI restriction sites—were amplified by PCR from a bacterial artificial chromosome clone (RP24-324H18; BACPAC Resources Center, Oakland Research Institute) that contained the mouse Sam68 gene; then, the amplified segments were sequentially inserted into the pGKneoF2L2DTA vector via restriction-enzyme digestion and ligation with T4 DNA ligase (NEB, M0202). After verification (by sequencing), the pGK-Sam68$^{floxEx5-8}$ targeting vector was linearized via restriction enzyme (Sca1) digestion, purified with a DNA purification kit (Qiagen), and transfected into C57BL/6N embryonic stem (ES) cells. Southern blotting (using a 5′ arm PCR Dig probe that had been synthesized with a commercially available kit [Roche, 11636090910]) and PCR analyses were conducted to confirm that the ES cells had incorporated the floxed Sam68-targeted locus by homologous recombination; then, the recombinant ES cells were microinjected into blastocysts

and implanted into pseudo-pregnant female mice at the Transgenic and Targeted Mutagenesis Laboratory in the Center for Genetic Medicine at Northwestern University. Chimeric off-spring were bred with WT C57BL/6J mice, and mice that harbored the Sam68$^{flox}$ allele (Sam68$^{f/w}$ mice) in their germline were identified via tail-snip PCR genotyping; then, the *Sam68*$^{f/w}$ C57Bl/6N mice were back-crossed to C57BL/6J mice for six generations to produce the mice used in this report. Sam68$^{f/f}$ mice were generated by interbreeding Sam68$^{f/w}$ heterozygote mice, and mice carrying the hepatocyte-specific knockout of Sam68 were produced by crossing Sam68$^{f/f}$ mice with Alb-cre mice (The Jackson Laboratory, No. 003574) to generate Alb-Cre$^{+}$; Sam68$^{f/f}$ (Sam68$^{LKO}$) mice and their Sam68$^{f/f}$ littermate controls.

To generate the Sam68$^{ΔN}$ transgenic mice, pcDNA3-HA-Sam68$^{ΔN}$ plasmids were linearized by digestion with restriction enzymes (NruI and StuI) to remove the vector sequence; then, the construct was purified and microinjected into the male pronuclei of C57BL6J oocytes at the Transgenic & Genetically Engineered Model Systems Core Facility of the University of Alabama at Birmingham (UAB). Three of the four founders were positive for the transgene (as determined via tail-snip PCR genotyping) and crossed with C57BL/6J mice to verify germline transmission; pups were evaluated via PCR genotyping and Western blotting of the liver, heart, and skeletal-muscle tissues with anti-HA.

The sequences of all primers and probes used for vector construction, PCR genotyping, and Southern blotting are reported in Supplementary Table 3, and the antibodies used for Western blotting are listed in Supplementary Table 2.

**GTT, PTT, GcTT, and ITT.** Mice were fasted overnight (16 h, GTT and PTT) or for 6 h (GcTT and ITT), and then glucose, sodium pyruvate, glucagon, or insulin was administered via intraperitoneal injection. Doses were 2 g/kg body weight glucose, 2 g/kg sodium pyruvate, 10 μg/kg glucagon, and 1 U/kg insulin for studies in non-diabetic models; 1 g/kg glucose, 1 g/kg sodium pyruvate, 10 μg/kg glucagon, and 2 U/kg insulin for studies in the HFD-STZ model; and 0.5 g/kg glucose, 0.5 g/kg sodium pyruvate, 5 μg/kg glucagon, and 2 U/kg insulin for *db/db* mice. Blood glucose levels were measured from tail bleeds 0, 15, 30, 45, 60, 90, and 120 min after injection, as previously described[59].

**Hyperinsulinemic-euglycemic clamping.** For catheter implantation surgery, mice were anesthetized with 2% inhaled isoflurane, and analgesia was provided via buprenorphine (0.05 mg/kg) and carprofen (2.5 mg/kg) injection before surgery, after surgery, and daily for 3 days during the recovery period. Home-made catheters were filled with heparinized-glycerol locking solution, implanted under the back skin of mice, and threaded into the left carotid artery for blood sampling and the right jugular vein for infusion. Five to 6 days after surgery, the mice were fasted for 5 h, and the catheters were externalized under isoflurane anesthesia and connected to syringe pumps (CMA 402, Harvard Apparatus, Holliston, MA; NE-300, New Era Pump Systems, Farmingdale, NY). Mice were awake, unhandled, and able to move freely in a plastic container during the study. The protocol consisted of a 120 min tracer equilibration period (from time, *t* = –120 to 0 min) beginning at 12:00 pm after a 4 h fasting period. For assessment of basal glucose turnover, [3-$^3$H]-Glucose (Perkin Elmer, Boston, MA) was delivered at *t* = –120 min first as a bolus dose (5 μCi), and then via continuous infusion (0.05 μCi/min) for 2 h. The insulin clamp was initiated at *t* = 0 min via continuous infusion of human insulin (1.2 mU/kg per min; Humulin R; Eli Lilly, Indianapolis, IN) and maintained until *t* = 120 min, and the [3-$^3$H]-Glucose infusion was increased to 0.1 μCi/min to minimize the change in specific activity from the equilibration period. The specific activity varied by less than 10% from the average during the last 40 min of the clamping period, and the slope of specific activity over time did not differ significantly from the slope at *t* = 0. Blood glucose levels were measured every 10 min with a Bayer Contour Blood Glucose Meter (Ascensia Diabetes Care US, Inc, Parsippany, NJ), and euglycemia (100 mg/dL) was maintained by adjusting the rate of infusion for the 20% glucose solution. Blood samples (50 μL) were taken for assessments of glucose, insulin, and free fatty acid levels, and for glucose specific

activity in plasma, at $t = -5$, 90, 100, 110, and 120 min; red blood cells were collected from the samples via centrifugation and injected via an arterial catheter to prevent hematocrit deficit. At the end of the clamp experiment, the mice were sacrificed, and the livers were snap-frozen in liquid nitrogen for measurements of glycogen content. The insulin infusion rate was based on observations in pilot studies.

**Biochemical assays.** Deproteinized plasma samples (20 μL) during clamping were used for measurements of total glucose concentrations by using Glucose Assay kit (Cell Biolabs, San Diego, CA) and for determination of [3-$^3$H]-Glucose and $^3$H$_2$O as we previously described[60]. Radioactivity was measured with a multi-purpose Scintillation Counter LS6500 (BECKMAN COULTER), and plasma glucose specific activity (SA) was calculated from the ratio of plasma glucose radioactivity (dpm) to plasma glucose content (mg) multiplied by the ratio of chemical standard evaporated (CSE) to chemical recovered standard (CRS); then, the [3-$^3$H]-glucose infusion rate (GIR; dpm/kg per min) was calculated from the CSE, the glucose disposal rate (Rd; mg/kg per min) was calculated as the ratio of the GIR to the plasma glucose SA at the end of the basal period and during the final 30 min of clamping, and the hepatic glucose production rate (Endo Ra; mg/kg per min) was calculated by subtracting the steady-state GIR from Rd. The radioactivity of $^3$H in hepatic glycogen was determined by digesting tissue samples in KOH and precipitating glycogen with ethanol[61]. Glycogen synthesis rate was converted by dividing hepatic tracer glucose infusion rate (dpm/g liver) by plasma tracer glucose SA.

**Serum insulin and lipid measurement.** Serum insulin was measured using the Ultra Sensitivity Mouse Insulin ELISA Kit (Crystal Chem, #90080) according to the manufacturer's protocol. Serum triglyceride and free fatty acid levels were measured using Infinity Triglyceride Reagent (ThermoFisher, TR22421) and free fatty acid Quantification Assay Kit (Abcam, ab65341), respectively, following manufacturer's instructions.

**HFD-induced diabetes model.** Four-week-old Sam68$^{LKO}$ and Sam68$^{f/f}$ mice were fed an HFD (58% of energy from fat; D12331, Research Diet) for 3 months, fasted for 6 h, intraperitoneally injected with a single dose of streptozotocin (STZ) (100 mg/kg body weight; freshly dissolved in 0.1 M of sodium citrate pH 4.5), and then maintained on the HFD for one more month. Blood glucose measurements were performed on day 5 after STZ injection and at the end of the HFD feeding protocol, and mice that were hyperglycemic (3 h fasting blood glucose levels ≥250 mg/dL) at both time points were diagnosed with diabetes (i.e. insulin defective stage of T2D)[62] and used in subsequent experiments.

**Human liver samples.** Studies with human tissues were approved by the Institutional Review Board (IRB) for Human Use of the University of Alabama at Birmingham (Protocol #: IRB-300002079) and performed in compliance with the Belmont Report and Declaration of Helsinki. Liver samples were obtained via surgical biopsy from patients with or without diabetes. Informed consent was obtained from all the subjects, and patient characteristics are listed in Supplementary Table 1.

**Plasmid construction, siRNAs, and cell transfection.** The pcDNA3-Flag-CRTC2 and pCMV6-COP1-Myc-Flag plasmids were purchased from Addgene and OriGene Technologies, respectively. The full-length pcDNA3-HA-Sam68 vector and serial truncation vectors (pcDNA3-HA-Sam68$^{ΔN-Ter(1-157aa)}$, pcDNA3-HA-Sam68$^{ΔCK(257-259aa)}$, pcDNA3-HA-Sam68$^{ΔP3-P4(280-346aa)}$, and pcDNA3-HA-Sam68$^{ΔC-Ter(347-443aa)}$) were constructed by GenScript (Piscataway, NJ) and verified by sequencing. The pcDNA3-Flag-CRTC2$^{K628R}$ plasmid was a gift from Dr. Marc Montminy (Salk Institute for Biological Studies)[63]. The non-targeting siRNA (D-001810-01-05) and targeting human CRTC2 siRNA (5′-UGGUUUACAU-GUCGACUAA-3′, J-018947-05-0002) were purchased from Dharmacon. Plasmids and siRNAs were transfected into 293T or HepG2 cells by using Lipofectamine 3000 Transfection Reagent (Invitrogen, Inc.).

**Adenoviral vectors and adeno-associated viral vectors.** Adenoviral vector Ad-CRTC2$^{K628R}$ was a gift from Dr. Marc Montminy (Salk Institute for Biological Studies)[63], and Ad-Sam68 and Ad-GFP were generated by Vector Biolabs (U Penn, Malvern, PA); the virus was administered to mice by tail vein injection at $2.0 \times 10^9$ infection units (IFU) per mouse and was applied to cultured primary hepatocytes at a dose of $5 \times 10^6$ PFU (plaque-forming unit) per $1.0 \times 10^6$ cells, respectively. AAV vectors (AAV8-TBG-eGFP, AAV8-TBG-iCre, AAV8-GFP-murineSam68-shRNA, and AAV8-GFP-Scrmb-shRNA) were generated by Vector Biolabs (U Penn, Malvern, PA) and administered by tail vein injection at $5 \times 10^{11}$ genome copies (AAV8-TBG-eGFP or AAV8-TBG-iCre) or $1 \times 10^{12}$ genome copies (AAV8-GFP-murinSam68-shRNA or AAV8-GFP-Scrmb-shRNA) per mouse; transgene expression was evaluated 3 weeks after administration.

**Hepatocyte isolation.** Primary hepatocytes were isolated via a two-step perfusion procedure with liver perfusion media (Krebs−Ringer Biocarbonate Buffer, Sigma, K4002) and liver digest buffer (Krebs−Ringer Biocarbonate Buffer with 0.1–0.15% collagenase) as previously described[64]. After isolation, cells were cultured on collagen-coated plates in DMEM containing 4.5 g/liter glucose and supplemented with 10% FBS, and 1% penicillin and streptomycin. After 6 h of attachment, the medium was replaced, and the cells were incubated overnight before using in subsequent experiments.

**Western blotting.** For protein extraction, $1 \times 10^7$ cells or 100 mg of frozen tissue were homogenized in 1 mL RIPA lysis buffer (50 mM Tris-HCl pH 8.0, 1 mM EDTA, 1% Triton X-100, 0.1% SDS, 150 mM NaCl) that contains protease-inhibitor (Sigma, 4693132001) and phosphatase-inhibitor (Sigma, 4906837001) cocktail. Samples were incubated with agitation for 30 min at 4 °C and centrifuged at $13000 \times g$ for 10 min at 4 °C; then, the protein concentration in the supernatant was determined via bicinchoninic acid (BCA) assay (Pierce). For immunoblotting, proteins in the supernatant were denatured by heating at 95 °C for 10 min, separated by SDS-PAGE, and then transferred onto a polyvinylidene difluoride (PVDF) membrane (Bio-Rad). The membrane was incubated in 5% non-fat milk blocking buffer (tris-buffered saline [TBS]) for 1 h, incubated with the primary antibody in TBS-containing 3% bovine serum albumin (BSA) overnight at 4 °C, washed 3 times with TBST (0.5% Tween 20), incubated with secondary antibody, washed with TBST, and then developed with Enhanced Chemiluminescence Detection Reagents (ECL, Thermo Fisher). Protein signals were imaged with a Bio-Rad ChemiDoc System. Antibodies are listed in Supplementary Table 2.

**Co-immunoprecipitations (co-IP).** Immunoprecipitation (IP) was performed as previously described[25]. Briefly, cells were lysed in 1% NP40 buffer (50 mM Tris-HCl pH 8.0, 1% Triton X-100, 150 mM NaCl, 0.25% sodium deoxy cholate, and protease inhibitor cocktail), and samples were incubated with protein A/G plus agarose-conjugated antibodies overnight at 4 °C and washed; then, the immuno-precipitates were eluted by boiling for 10 min, and extracts were analyzed by Western Blotting. To evaluate CRTC2 ubiquitination, IP Lysis buffer was supplemented with a complete protease inhibitor cocktail (Thermo Fisher, 4311235), a deubiquitin enzyme inhibitor PR-619 (30 uM, EMD Millipore), and 1,10-Phenanthroline (5 mM, EMD Millipore).

**Quantitative real-time polymerase chain reaction (qRT-PCR).** Total RNA was isolated with TRIzol Reagent and reverse transcribed into cDNA with Reverse Transcription Reagents (Applied Biosystems); then, tissue mRNA levels were determined by qPCR (ABI3000; Applied Biosystems) with SYBR Green Real-Time PCR Master Mix (Applied Biosystems). Duplicate reactions were performed for each sample, and the relative mRNA expression level for each gene was calculated via the 2(−ΔΔCt) method and normalized to β-actin. Primers are listed in Supplementary Table 3.

**Chromatin immunoprecipitation (ChIP).** ChIP experiments were performed using a ChIP assay kit (EMD Millipore) as directed by the manufacturer's instructions. Briefly, cells were fixed in 1% formaldehyde for 10 min at room temperature; then, the cells were quenched by adding glycine to a final concentration of 125 mM, washed twice with PBS containing a protease inhibitor cocktail (1 mM PMSF, 1 μg/ml aprotinin, and 1 μg/ml pepstatin A), pelleted, resuspended in 200 μL ChIP lysis buffer (1% SDS, 10 mM EDTA, and 50 mM Tris, pH 8.1), and sonicated in an ultrasonic processor (Sonicator 3000; Misonix) to shear the DNA into ~500-bp segments. Samples were precleared with protein A agarose beads and then immunoprecipitated with CRTC2 antibody (sc-271912; Santa Cruz Biotechnology, Inc.) or rabbit IgG at 4 °C overnight. Immunoprecipitated DNA complexes were eluted from the agarose beads twice by adding elution buffer (1% SDS, 0.1 M NaHCO3, pH 8.0) at room temperature for 15 min, and cross-linking was reverse by heating at 65 °C for 2 h. The immunoprecipitated DNA was analyzed by quantitative real-time PCR with primers for regions encompassing the CRE motif in the promoters of PGC-1α, PEPCK, and G6Pase. Primers are listed in Supplementary Table 3.

**Hepatic glucose production.** Primary hepatocytes were seeded into six-well plates ($1 \times 10^6$ cells per well), cultured overnight in DMEM with 10% FBS, washed three times with PBS, and then incubated for 4 h in glucose production buffer (consisting of glucose-free DMEM pH 7.4 without phenol red and supplemented with 20 mM sodium lactate, 2 mM sodium pyruvate, and 1 mM glycerol) containing 100 nM glucagon, 10 μM forskolin, or 100 μM Bt2-cAMP; then, 0.5 mL medium was collected, and the glucose concentration was measured with a colorimetric glucose assay kit (Eton Bioscience, Inc). Readings were normalized to the total protein content in whole-cell lysates.

**Cytoplasmic and nuclear fractionation.** Cytoplasmic and nuclear fractions were isolated with a commercially available kit (78835; Thermo Fisher Scientific) as directed by the manufacturer's instructions. For assessments of glucagon-induced nuclear translocation in the liver, mice were fasted for 4 h and intraperitoneally injected with glucagon (30 μg/kg body weight) via the inferior cava; 10 min later, the liver was collected, and cytoplasmic and nuclear proteins were isolated.

**Analysis of in vivo insulin signaling**. Mice were fasted for 16 h and intraperitoneally injected with insulin or saline. Twenty minutes later, mice were euthanized, and the liver tissues were quickly excised, snap-frozen in liquid nitrogen, and stored at −80 °C until use. For evaluation of insulin signaling, tissues were homogenized in RIPA buffer containing a protease- and phosphatase-inhibitor cocktail (Sigma); then, tissue proteins were extracted and evaluated via Western blotting with primary antibodies against Ser-473-AKT, Thr-308-AKT, and total AKT (Supplementary Table 2) as previously described[31].

**PKA activity assay**. Cells were starved in FBS-free DMEM for 4 h, treated with 100 nM glucagon, 10 μM forskolin, or 100 μM Bt2-cAMP for 30 min, and then lysed in cell lysis buffer (tris-based, pH 8 buffer with 1% NP-40) containing protease-inhibitors and phosphatase-inhibitors cocktail (Sigma). PKA activity was determined with a PKA Colorimetric Activity Kit (Invitrogen, #EIAPKA) as directed by the manufacturer's protocol.

**Targeting protein domain prediction by text pattern search and complementary hydropathy**. Mouse Sam68 and CRTC2 sequences were obtained from UniProt (accession Q60749-1, and accession Q3U182-1, respectively). Combined text pattern search method[65] and the hydropathic analysis method[66] were used to predict potential Sam68 C-terminal domain (P5, "YY" and NLS region) binding sites in CRTC2. The sequences of the proline-rich region of P5 (356-363 aa), the tyrosine-rich "YY" region (366-411 aa), and nuclear localization signal (420-443 aa) region in the C-terminal of Sam68 were separately converted into binary (+ or −) hydrophobicity maps based on the sign of each amino acid's hydrophobicity by the Kyte and Doolittle scale. Hydrophobicity map for each potential binding motif in Sam68 in both forward and reverse orientations was scanned across a hydrophobicity map for the whole protein sequence of CRTC2. The percentage of match was calculated by dividing complementary (+/−) pairs to all matched pairs. The degree of complementary hydropathy was calculated (see Eq. 1[66]) based on the Kyte Doolittle hydropathy index[67].

$$C = \frac{\sum_{i=1}^{L}|H(i) - H'(i)|}{L*9}$$

In Eq. 1, $C$ is the degree of complementary hydropathy, $H(i)$ and $H'(i)$ are the hydropathy indices of the amino acids in the motif and target sequences respectively at position $i$, and $L$ is the length of the protein fragment. The degree of complementary hydropathy can range from 0 to 1.0. Only ≥75% percentage of the match and ≥0.42 degree of complementary hydropathy were considered as a high possibility for protein interaction in this study. The hydropathy plot was generated according to a hydropathic score of each amino acid.

**Statistics and reproducibility**. Data are presented as mean ± s.e.m. Statistical significance between two groups was evaluated via the unpaired two-tailed Student's $t$-test and among three or more groups via one-way or two-way analysis of variance (ANOVA) with one or two independent variables. GraphPad Prism8 Software was used for statistical analysis. A $p$-value of less than 0.05 was considered significant. All the representative Western blotting and measurements of glucose production from cultured primary hepatocytes presented in this article were repeated at least twice with similar results.

**Reporting summary**. Further information on research design is available in the Nature Research Reporting Summary linked to this article.

## Data availability

All data are included in this published article and its Supplementary information files. Source data are provided with this paper.

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

## Acknowledgements

We thank Dr. Marc Montminy (Salk Institute for Biological Studies) for generously providing pcDNA3-Flag-CRTC2$^{K628R}$ plasmid and CRTC2$^{K628R}$ expressing adenoviral vector. We thank Mr. Rory A Greer for assistance in text pattern search and hydropathy analysis for protein binding site prediction. We thank the UAB Small Animal Phenotyping and Glycemic Clamp Cores supported by the National Institutes of Health Nutrition Obesity Research Centers (P30-DK-056336). This work was supported by the National Institute of Health (R01 Grants# HL113541, HL130052, HL131110, and HL138990 to G. Q.; HL142291 to H. Q and G. Q.; 1R01DK112934 to K. M. H); American Diabetes Association (Grant# 1-15-BS-148 to G.Q.); American Heart Association (Grant# 19TPA34910227 to G. Q.; 19CDA34630052 to A. Q.; 18POST34070088 to S. X; and 18PRE34080358 to E. Z.).

## Author contributions

A.Q. and G.Q. conceptualized the study, interpreted data, and wrote the paper. A.Q. performed most of the experiments and statistical analyses. J.Z., S.X., W.M., C.B., T.K., B.Y., J.D., LY., and Y.S. performed experiments and analyzed data. E.Z. made important editing. Y.M. and S.R. contributed key reagents. Y.S., Y.M., S.R., C.Z., H.Q., K.H., and J.Z. made intellectual contributions and assisted in data interpretation.

## Competing interests

The authors declare no competing interests.
