## [Peer Review File · Nature Communications]

Reviewers' Comments:

Reviewer #1:

Remarks to the Author:

The study by Qiao and co-authors investigates the role played by Sam68 in hepatic gluconeogenesis. By using elegant mouse models, the authors uncover a direct role of Sam68 as adaptor protein in this process. In particular, they propose that Sam68 binds to the gluconeogenic transcription factor CRTC2 and stabilizes it. Conversely, ablation of Sam68 lowers the expression levels of CRTC2, reduces gluconeogenesis and glucose blood levels while increasing insulin sensitivity. Importantly, Sam68 and CRTC2 expression was upregulated in the liver of both patients with diabetes and in two diabetic mouse models (HFD-fed and db/db). Depletion of Sam68 expression in the liver strongly ameliorated the hyperglycemic phenotype of diabetic mice. These findings depict a new gluconeogenic pathway and suggest that targeting Sam68 function/expression in the liver may represent a therapeutic strategy for Type 2 diabetes. Nevertheless, some key points should be addressed to fully support the conclusions of this study.

Main Points:

Figure 1: the reduced levels of glucose in the blood was already reported for Sam68 knockout mice (Huot et al., Mol Cell 2012). This information should be provided in the text and discussed for congruency or differences with the present study.

Figure 2: the authors should also evaluate the expression of CRTC1 and CRTC3 protein levels in Sam68KO hepatocytes and/or in liver of Sam68LKO mice.

Figure 3: to confirm that CRTC2 protein degradation is the limiting step of gluconeogenesis in Sam68LKO mice, the authors should compare the effect of stable expression of wild type and degradation-resistant variants of CRTC2 (Ad-CRTC2wt vs Ad-CRTC2K628R). This experiment would demonstrate that the reduced gluconeogenesis, due to depletion of Sam68 expression, occurs only when wild type CRTC2 is over-expressed and then degraded, and not in presence of the degradation-resistant variant of CRTC2. On the other hand, if the wt is sufficient, the experiment would not prove the role of degradation by the proteasome.

Figure 4A: the authors claim that there is a dramatic increase in the nuclear level of Sam68 following glucagon treatment, which appears an overstatement. Quantification of the nuclear translocation of Sam68 upon glucagon treatment should be provided (densitometric analysis of at least 3 replicates performed for this experiment), as well as of the comparison with the western blot analysis for the total protein level of Sam68 in control or glucagon treated cells. Importantly, nuclear and cytosolic samples should be loaded and stained on the same gel, to determine the cytoplasmic contamination (actin staining) of nuclei or the nuclear contamination (Lamin A/C staining) of the samples. Moreover, this type of representation would allow to determine the relative proportion of cytoplasmic versus nuclear Sam68 in the samples and is a more accurate way to quantify translocations.

Figure 4B: a proper control is the reverse co-immunoprecipitation. It is expected that immunoprecipitation of Sam68 would bring down CRTC2 only in Sam68 wild type cells. This experiment should be done. Also, is the interaction occurring in the nucleus or cytoplasm? This is relevant, as the delta-C mutant used in panel D is likely to be entirely cytoplasmic (the C-terminal region contains the NLS of Sam68). The authors should perform co-IP in cytoplasmic and nuclear extracts and determine the subcellular localization of the Sam68 mutants to draw conclusions on the domains required for the interaction. In the discussion, the authors propose that translocates from the cytoplasm to the nucleus upon glucagon stimulation. However, Sam68 is predominantly nuclear in most cells. The authors should evaluate the subcellular localization of Sam68 by immunofluorescence analysis to corroborate the subcellular fractionations, which may present artifacts.

Figure 4 C,D: in the co-immunoprecipitation experiments with tagged-protein, immunoprecipitation with the tag-specific antibody of cells transfected with the tag-vector only represents a more appropriate control than unrelated IgG control.

Figure 4H: the increase in ubiquitination of CRTC2 in Sam68 knockout cells is not evident. More quantitative assays should be provided to corroborate the results.

Figure 4I: it is not clear if the western blot analysis for Sam68 and CRTC2 were performed using antibodies specific for the endogenous or the tagged proteins, especially for CRTC2. Shouldn't both endogenous and tagged CRTC2 proteins be visible in the blot if an anti-CRTC2 antibody was used? Please clarify.

Figure 4H: quantification of the increase in polyubiquitination in Sam68 KO in at least 3 experiments should be performed, as the increase shown appears quite mild, if not negligible.

Figure 4J: as mentioned above, before drawing conclusions, the authors should determine where the Sam68/CRTC2 and CRTC2/COP1 interactions occur. If the compartment is the nucleus, the results may be affected by the subcellular localization of the mutants. In vitro experiments with purified proteins are more appropriate to identify proteins domains involved in direct interactions. The intracellular site of interaction between CRTC2 and Sam68 or COP1 should be determined. If the authors narrow down the interaction to the P5 motif of Sam68, site-specific mutations of these prolines would be a more appropriate model to corroborate their hypothesis.

Figure 5: the authors do not justify the choice of the Sam68 deltaN-Tg mouse model. If the Sam68 delta-C mutant was more deficient in the interaction, why they used the delta-N? It is unclear how the delta-N mutant of Sam68 loses the ability to protect CRTC2 from degradation and why it should act as dominant negative if it is not lacking the CRTC2 binding domain. Is the interaction between CRTC2 and COP1 affected? How does the N-terminal region of Sam68 contributes to this process if the protein interacts with CRTC2 through its C-terminal domain? As pointed for the experiments in Figure 4, a more detailed investigation of the interactions in play in this process is required. For instance, co-immunoprecipitation experiments between COP1 and CRCT2 in presence of the different Sam68 mutants could clarify if mutants with similar interaction properties with CRTC2 differentially affect its interaction with COPI and if these proteins form a ternary complex. As mentioned above, the subcellular localization of Sam68 mutants should be verified and taken into account.

Figure 6: as mentioned above, nucleo-cytoplasmic fractionations should be loaded and stained in the same gel to be evaluated.

Minor Points:

Figure 4: a schematic representation of Sam68 domains and Sam68 mutants used in the experiments could help the comprehension of the data. Moreover, some controls in the immunoprecipitation experiments are missing (i.e. IgG in panel b, d and h).

Figure 6: how is Sam68 protein up-regulated in diabetic models? Sam68 protein is very stable and its expression level are usually determined by transcriptional regulation. Have the authors investigated this issue? What type of post-transcriptional regulation is involved?

Third paragraph of the Introduction: when referring to the spermatogenic defects of Sam68 knockout mice, the authors should cite Paronetto et al., J Cell Biol 2009, where the defect was initially reported and characterized. In Li et al., Fertility and Sterility 2014, the authors report a reduced level of Sam68 in human patients with spermatogenic defects, not a direct role of Sam68 in mouse spermatogenesis. Similarly, few lines below, when they refer to the "increase in adipose thermogenesis and improved systemic glucose profile on high fat diet", the authors should also cite reference 30 (Huot et al., Mol Cell 2012), which anticipated some of these phenotypes with respect to the cited reference (Zhou et al., J Endocrinology 2015).

Reviewer #2:

Remarks to the Author:

In this study Qiao et al. identified a previously unrecognized role of Sam68 in promoting hepatic gluconeogenesis by binding and stabilizing CRT2. This study provides new insight into glucose production and suggests that targeting Sam68 may normalize glycemia in patients with diabetes. The data strongly support most of the conclusions. The conclusions will be further strengthened with additional experiments or discussion.

Major points:

1. Are protein levels of PGC-1 α , PEPCK and G6Pase increased during fasting (Fig. 2b)? It is better to clearly show the results with additional statistical data.
2. CREB should not be present in cytosolic fractions (Fig. S3c).
3. Since the authors think the amino acids from 356 to 363 of Sam68 and 77-84 of CRT2 are important for Sam68: CRT2 binding, the authors can try to define which amino acid(s) is really critical for the association by mutant screening.
4. Compared to WT Sam68, Sam68 Δ C loses its association with CRT2 while Sam68 Δ N still has a weak association with CRT2 (Fig. 4d). Why do the authors choose Sam68 Δ N for the subsequent experiments?

Minor points:

1. The labelling of Ub-CRT2 is incorrect for input (Fig. 4h).
2. Fig. 6I is redundant to Fig. 6H.
3. Was there no difference of blood glucose at time 0 min (Fig. 1r)?

Reviewer #3:

Remarks to the Author:

This work from Dr. Qin group identified Sam68 as a new regulator of hepatic gluconeogenesis. They discovered that both global and hepatic deletion of Sam68 reduce blood glucose and impair glucagon-induced expression of gluconeogenic genes. Sam68 promotes hepatic gluconeogenesis possibly via its interaction with CRT2 and enhancing its protein stability. Lastly they found that hepatic Sam68 expression is upregulated in patients with diabetes and mouse models of diabetes, indicating elevation of Sam68 could drive hyperglycemia in the context of diabetes. Overall this is a very interesting finding that is supported by strong animal data and innovative molecular basis. However there are some concerns that need to be addressed in the current version of manuscript.

Major concerns:

1. Lack direct evidence supporting Sam68 promotes hepatic gluconeogenesis via CRT2. A good experiment should be done with over-expression of Sam68 in either WT or Crt2 $^{-/-}$ hepatocyte.
2. Other important factors for gluconeogenesis are not examined in the liver of Sam68-ko mice, including FOXO1, HNF4a, GR.
3. Sam68 itself is a RNA-binding protein. How exactly Sam68 promotes CRT2 protein stability remains unaddressed in the current manuscript? Does Sam68 compete binding to CRT2 with putative E3 ligase?
4. Hepatic CRT2 is also found to be important for lipid metabolism via SREBP1. The authors should provide lipid phenotype in Sam68 mouse models. It is possible that glucose phenotype is secondary to altered lipid metabolism in Sam68 ko mice.

Minor concerns:

1. Figure 2. The glucagon dose for treating mouse hepatocyte was 100nM, which is too high for in vitro study. The lower dose of glucagon should be tested.
2. Poor quality of some immunoblotting with over-exposed images, including Figure 3a, 4a, 4h, 6a, 6b, 6e and 6c.
3. Figure 4, author should test over-expression of Sam68 on CRT2 protein half-life.
4. Figure 6, how about body weight after administrating shSam68 in ob/ob mice? Also lipid profile in these cohorts?
5. How is Sam68 level induced in diabetic conditions? This should be discussed or at least tested using in vitro experiments.

Response to Reviewers

Reviewer #1

Major comments

[a] *Figure 1: the reduced levels of glucose in the blood was already reported for Sam68 knockout mice (Huot et al., Mol Cell 2012). This information should be provided in the text and discussed for congruency or differences with the present study.*

Response

We thank the reviewer for the suggestion, and have now added the following sentence in paragraph 1 of the Discussion section, as follows:

“While an improved systemic glucose profile in HFD-fed Sam68^{-/-} mice was previously reported by us ^{1,2}, the results presented here are the first to show that Sam68 has a key regulatory role in gluconeogenesis and blood glucose maintenance.”

[b] *Figure 2: the authors should also evaluate the expression of CRTC1 and CRTC3 protein levels in Sam68KO hepatocytes and/or in liver of Sam68^{LKO} mice.*

Response

We appreciate this suggestion and have analyzed the levels of CRTC1 and CRTC3 proteins in the liver of mice; Sam68^{LKO} mice and Sam68^{fl/fl} littermates expressed similar amounts of each isoforms under feeding or fasting condition (Supplementary Figure 3g of our revised manuscript).

[c] *Figure 3: to confirm that CRTC2 protein degradation is the limiting step of gluconeogenesis in Sam68^{LKO} mice, the authors should compare the effect of stable expression of wild type and degradation-resistant variants of CRTC2 (Ad-CRTC2^{wt} vs Ad-CRTC2^{K628R}). This experiment would demonstrate that the reduced gluconeogenesis, due to depletion of Sam68 expression, occurs only when wild type CRTC2 is over-expressed and then degraded, and not in presence of the degradation-resistant variant of CRTC2. On the other hand, if the wt is sufficient, the experiment would not prove the role of degradation by the proteasome.*

Response

We appreciate the reviewer's suggestion. During the execution of this experiment, we indeed considered using Ad-CRTC2^{WT} as a control for Ad-CRTC2^{K628R}, but encountered the limitation of this Ad virus system, in which CRTC2^{WT} overexpression via Ad-CRTC2^{WT} overwhelms the ubiquitin-proteasome system, leading to partially degradation, thus uncertainty to what extent the reduced CRTC2^{WT} level was due to degradation vs. due to lowered Ad-CRTC2^{WT} transduction efficiency; we will develop an expression-level controlled and hepatic-specific system for both CRTC2^{WT} and CRTC2^{K628R} expression and conduct similar experiments in our future studies. Importantly, CRTC2 is a well-recognized master limiting regulator of gluconeogenesis. A modest elevation of CRTC2 increased blood glucose levels ^{3,4}, and a reduction of CRTC2 protein level by 50% leads to hypoglycemia and increased insulin sensitivity ⁵; and our results indicate that in Sam68-knockout liver and hepatocytes, the CRTC2 protein level is reduced by

>50% (Figures 3a, 3i, 4f-h, 6k, and Supplementary Figures 3a-b, 3g). Thus we believe our conclusion that the lowered gluconeogenesis and hypoglycemia in Sam68-knockout mice is mediated by CRTC2 protein degradation is fully supported.

[d] Figure 4A: the authors claim that there is a dramatic increase in the nuclear level of Sam68 following glucagon treatment, which appears an overstatement. Quantification of the nuclear translocation of Sam68 upon glucagon treatment should be provided (densitometric analysis of at least 3 replicates performed for this experiment), as well as of the comparison with the western blot analysis for the total protein level of Sam68 in control or glucagon treated cells. Importantly, nuclear and cytosolic samples should be loaded and stained on the same gel, to determine the cytoplasmic contamination (actin staining) of nuclei or the nuclear contamination (Lamin A/C staining) of the samples. Moreover, this type of representation would allow to determine the relative proportion of cytoplasmic versus nuclear Sam68 in the samples and is a more accurate way to quantify translocations.

Response

We have repeated these experiments in mouse liver as well as in cultured hepatocytes by completely following the reviewer's suggestions. The results confirm our original observation that glucagon treatment induces Sam68 nuclear translocation (Figure 4a and Supplementary Figure 4a of our revised manuscript).

[e] Figure 4B: a proper control is the reverse co-immunoprecipitation. It is expected that immunoprecipitation of Sam68 would bring down CRTC2 only in Sam68 wild type cells. This experiment should be done. Also, is the interaction occurring in the nucleus or cytoplasm? This is relevant, as the delta-C mutant used in panel D is likely to be entirely cytoplasmic (the C terminal region contains the NLS of Sam68). The authors should perform co-IP in cytoplasmic and nuclear extracts and determine the subcellular localization of the Sam68 mutants to draw conclusions on the domains required for the interaction. In the discussion, the authors propose that translocates from the cytoplasm to the nucleus upon glucagon stimulation. However, Sam68 is predominantly nuclear in most cells. The authors should evaluate the subcellular localization of Sam68 by immunofluorescence analysis to corroborate the subcellular fractionations, which may present artifacts.

Response

We appreciate the reviewer's suggestions and have performed reverse co-immunoprecipitations using the fractionated cellular extracts; the results confirmed Sam68-CRTC2 interaction in primary hepatocytes and further revealed that the interaction occurs in both nucleus and cytoplasm (Supplement Figure 4d of our revised manuscript). We feel that the results of our new experiments on glucagon-induced Sam68 nuclear localization (Figure 4a and Supplementary Figure 4a) and on Sam68-CRTC2 interactions (Supplement Figure 4d) are very clear and would avoid the need for immunofluorescence staining, which is less quantitative and in this particular case, complicated by the much larger quantity of Sam68 protein in the nucleus vs. cytoplasm.

[f] Figure 4 C, D: in the co-immunoprecipitation experiments with tagged-protein, immunoprecipitation with the tag-specific antibody of cells transfected with the tag-vector only represents a more appropriate control than unrelated IgG control.

Response

To address the reviewer's concern about potential non-specific identifications by anti-HA, we performed co-immunoprecipitation experiments using anti-HA antibody in cells transfected with an HA-tag empty vector or Flag-tagged CRTC2, then detected the presence of CRTC2 protein in the immunoprecipitates using anti-Flag antibody; no Flag-CRTC2 band was present in the anti-HA immunoprecipitates (Supplementary Figure 4j of the revised manuscript). Thus, our identification of Sam68-CRTC2 interaction in this report is specific.

[g] Figure 4H: the increase in ubiquitination of CRTC2 in Sam68 knockout cells is not evident. More quantitative assays should be provided to corroborate the results. Quantify.

Response

We have repeated these experiments with three replicates; the representative image has been significantly improved, and quantitative data demonstrate a significantly increased CRTC2 ubiquitination in Sam68 knockout cells vs. WT cells (the relabeled Figure 4i of our revised manuscript).

[h] Figure 4I: it is not clear if the western blot analysis for Sam68 and CRTC2 were performed using antibodies specific for the endogenous or the tagged proteins, especially for CRTC2. Shouldn't both endogenous and tagged CRTC2 proteins be visible in the blot if an anti-CRTC2 antibody was used? Please clarify.

Response

We apologize for the lack of clarity. We used anti-CRTC2 (detection of both endogenous and transfected CRTC2) and anti-HA (detection of the transfected Sam68) in these experiments. The detailed antibody information are now indicated on the left side of the blots (Figure 4j of our revised manuscript).

[i] Figure 4H: quantification of the increase in polyubiquitination in Sam68 KO in at least 3 experiments should be performed, as the increase shown appears quite mild, if not negligible.

Response

Please refer to our response to Reviewer# 1/Comment [g].

[j] Figure 4J: as mentioned above, before drawing conclusions, the authors should determine where the Sam68/CRTC2 and CRTC2/COPI interactions occur. If the compartment is the nucleus, the results may be affected by the subcellular localization of the mutants. In vitro experiments with purified proteins are more appropriate to identify proteins domains involved in direct interactions. The intracellular site of interaction between CRTC2 and Sam68 or COPI should be determined. If the authors narrow down the interaction to the P5 motif of Sam68, site-specific mutations of these prolines would be a more appropriate model to corroborate their hypothesis.

Response

As noted in our response to Reviewer# 1/Comment [e], Sam68-CRTC2 interaction occurs in both nucleus and cytoplasm (Supplement Figure 4d of our revised manuscript). To confirm the

role of the P5 motif, we generated P5 truncation mutant (HA-Sam68^{ΔP5}), in which 356-363 amino acid (PLPPTPAP) has been deleted, and performed co-immunoprecipitation analyses; Flag-CRTC2 failed to bind HA-Sam68^{ΔP5}, confirming that P5 motif is essential for the Sam68-CRTC2 interaction (Supplemental Figure 4i of our revised manuscript).

[k] Figure 5: the authors do not justify the choice of the Sam68 deltaN-Tg mouse model. If the Sam68 delta-C mutant was more deficient in the interaction, why they used the delta-N? It is unclear how the delta-N mutant of Sam68 loses the ability to protect CRTC2 from degradation and why it should act as dominant negative if it is not lacking the CRTC2 binding domain. Is the interaction between CRTC2 and COP1 affected? How does the N-terminal region of Sam68 contributes to this process if the protein interacts with CRTC2 through its C-terminal domain? As pointed for the experiments in Figure 4, a more detailed investigation of the interactions in play in this process is required. For instance, co-immunoprecipitation experiments between COP1 and CRCT2 in presence of the different Sam68 mutants could clarify if mutants with similar interaction properties with CRTC2 differentially affect its interaction with COP1 and if these proteins form a ternary complex. As mentioned above, the subcellular localization of Sam68 mutants should be verified and taken into account.

Response

We apologize for the lack of clarity about the rationale for choosing Sam68^{ΔN}. Based on our co-IP and hydropathy analyses, Sam68 C-ter (via P5) interacts with the N-ter of CRTC2, which likely triggers a conformational change to allow Sam68 N-ter to interfere with COP1-mediated CRTC2 ubiquitination and degradation. To address the reviewer's concern and further strengthen this assertion, we have performed additional experiments and found that overexpression of Sam68^{ΔN} dose-dependently suppresses COP1-mediated CRTC2 degradation (Supplementary Figure 4f of our revised manuscript), and that Sam68^{ΔC} alone is not sufficient to inhibit COP1-mediated CRTC2 degradation unless at extremely high Sam68^{ΔC} concentration (Figure. 4k and Supplemental Figure.4f of our revised manuscript). Thus Sam68^{ΔN} plays a dominant negative role in COP1-mediated CRTC2 degradation *in vitro*, and we chose Sam68^{ΔN} transgenic mice for those *in vivo* experiments.

[l] Figure 6: as mentioned above, nucleo-cytoplasmic fractionations should be loaded and stained in the same gel to be evaluated.

Response

We have repeated these experiments following reviewer's suggestions (Figure 6a and 6b of the revised manuscript).

Minor Points:

[m] Figure 4: a schematic representation of Sam68 domains and Sam68 mutants used in the experiments could help the comprehension of the data. Moreover, some controls in the immunoprecipitation experiments are missing (i.e. IgG in panel b, d and h).

Response

We have now included a schema of the Sam68 domains and Sam68 mutants (Figure 4d of our revised manuscript). IgG controls are also provided in these three panels; please note that panels 4d and 4h have been relabeled as panels 4e and 4i, respectively.

[n] *Figure 6: how is Sam68 protein up-regulated in diabetic models? Sam68 protein is very stable and its expression level are usually determined by transcriptional regulation. Have the authors investigated this issue? What type of post-transcriptional regulation is involved?*

Response

We thank the reviewer for the insightful questions and have addressed these questions by including the following statement in the Discussion section (paragraph 4 of our revised manuscript).

“We observed that hepatic Sam68 protein expression is significantly upregulated in diabetic mouse models and in human subjects with diabetes, that glucagon signaling promotes Sam68 translocation from cytoplasm to nucleus, and that Sam68 interacts with CRTC2 both in the nucleus and cytoplasm. Evidence from other laboratories suggest that insulin induces nuclear export of Sam68 in rat adipocytes ^{6,7}, and a variety of posttranscriptional modifications, including tyrosine phosphorylation ⁸, serine/threonine phosphorylation ⁹, acetylation ¹⁰, methylation ¹¹ and SUMOylation ¹², are associated with Sam68 localization and stability. Thus, it is likely that under diabetic conditions, deregulated metabolic signaling (e.g., glucagon and insulin) result in skewed posttranscriptional modifications and subcellular trafficking of Sam68 to increase CRTC2 stability, and studies to identify these site-specific modifications may provide additional mechanistic insights and therapeutic targets of T2D.”

[o] *Third paragraph of the Introduction: when referring to the spermatogenic defects of Sam68 knockout mice, the authors should cite Paronetto et al., J Cell Biol 2009, where the defect was initially reported and characterized. In Li et al., Fertility and Sterility 2014, the authors report a reduced level of Sam68 in human patients with spermatogenic defects, not a direct role of Sam68 in mouse spermatogenesis. Similarly, few lines below, when they refer to the “increase in adipose thermogenesis and improved systemic glucose profile on high fat diet”, the authors should also cite reference 30 (Huot et al., Mol Cell 2012), which anticipated some of these phenotypes with respect to the cited reference (Zhou et al., J Endocrinology 2015).*

Response

We sincerely apologize for the omissions in the original draft, and have now included these two articles as suggested (paragraph 3 of the Introduction section).

Reviewer #2

Major points:

1. Are protein levels of PGC-1 α , PEPCK and G6Pase increased during fasting (Fig. 2b)? It is better to clearly show the results with additional statistical data.

Response

We appreciate this important question. We have repeated these experiments and performed quantitative analyses of these proteins in the fasted liver; the PGC-1 α , PEPCK, and G6Pase proteins were upregulated during fasting, however their levels were significantly lower in Sam68^{LKO} mice than in Sam68^{f/f} mice under both feeding and fasting conditions (Figure 2b of our revised manuscript).

2. CREB should not be present in cytosolic fractions (Fig. S3c).**Response**

We thank the reviewer for pointing out. In our original submission, this might have occurred due to cytoplasm fractions being contaminated with small amount of nuclear fractions. For our revised submission, we repeated this experiment making sure no cross-contamination by loading nuclear and cytoplasmic fractionations on the same gel, as also suggested by the Reviewer #1; no CREB is present in the cytosolic fractions (Supplementary Figure 3c of our revised manuscript).

3. Since the authors think the amino acids from 356 to 363 of Sam68 and 77-84 of CRTC2 are important for Sam68:CRTC2 binding, the authors can try to define which amino acid(s) is really critical for the association by mutant screening.**Response**

We appreciate this suggestion. As noted in our response to Reviewer#1/Comment [j], we generated a P5 motif truncation mutant (HA-Sam68^{ΔP5}), in which 356-363 amino acids (PLPPTPAP) were deleted, and co-immunoprecipitation experiments confirmed that Sam68 P5 is essential for Sam68-CRTC2 interaction (Supplemental Figure 4j of our revised manuscript).

4. Compared to WT Sam68, Sam68deltaC loses its association with CRTC2 while Sam68deltaN still has a weak association with CRTC2 (Fig. 4d). Why do the authors choose Sam68deltaN for the subsequent experiments?**Response**

We appreciate this important question and apologize for the lack of clarity about the rationale for choosing Sam68^{ΔN}. Please refer to our detailed response to Reviewer #1/Comment [k].

Minor points:**1. The labelling of Ub-CRTC2 is incorrect for input (Fig. 4h).****Response**

We apologized for the mislabeling. This has been corrected (Figure 4i of our revised manuscript).

2. Fig. 6l is redundant to Fig. 6H.**Response**

Thank you for the suggestion. We have now removed Figure 6l and reordered Figure 6 in the

revised submission.

3. Was there no difference of blood glucose at time 0 min (Fig. 1r)?

Response

There was difference ($p < 0.05$). We apologize for the accidental omission. The p value is now indicated.

Reviewer #3

Major concerns:

1. Lack direct evidence supporting Sam68 promotes hepatic gluconeogenesis via CRT2. A good experiment should be done with over-expression of Sam68 in either WT or Crtc2^{-/-} hepatocyte.

Response

We appreciate this insightful suggestion and have performed additional experiment to overexpress Sam68 in HepG2 cells; Sam68 overexpression dramatically increased glucagon-induced glucose production and gluconeogenic gene expression in WT HepG2 cells but not in the HepG2 cells with CRT2 knockdown (Supplemental Figure 3o-q), which further supports our conclusion that Sam68 promotes hepatic gluconeogenesis via CRT2.

2. Other important factors for gluconeogenesis are not examined in the liver of Sam68-ko mice, including FOXO1, HNF4a, GR.

Response

We appreciate this important comment and have performed additional Western blotting analyses that examined the expression of FOXO1, HNF4 α and GR proteins in the liver of mice under both feeding and fasting conditions; the levels of all three proteins were similar between Sam68^{f/f} and Sam68^{LKO} mice (Figure 2b and 2g of our revised manuscript).

3. Sam68 itself is a RNA-binding protein. How exactly Sam68 promotes CRT2 protein stability remains unaddressed in the current manuscript? Does Sam68 compete binding to CRT2 with putative E3 ligase?

Response

We appreciate the reviewer's questions. Sam68 is an RNA binding protein but also an adaptor protein that has been shown to interact with many proteins to regulate cellular signaling and functions^{8,13-21}. Data from our mechanistic studies support that Sam68 (via P5 motif at C-terminus) interacts with CRT2 (likely N-terminus), thus proximity of Sam68 N-terminus, to elicit CRT2 conformational changes, which inhibits E3 ubiquitin ligase COP1-mediated CRT2 ubiquitination at K628 and proteasome degradation (Figure 4e, 4j-k and Supplemental Figure 4f). It is known that COP1 can bind to the amino acids 209–215 (EMDPKVP) of CRT2 to catalyze K628 ubiquitination; whether Sam68-CRT2 interaction directly interferes COP1-

CRTC2 binding and if additional interacting proteins also participate in the regulation of CRTC2 K628 ubiquitination will be investigated in our future studies.

4. Hepatic CRTC2 is also found to be important for lipid metabolism via SREBP1. The authors should provide lipid phenotype in Sam68 mouse models. It is possible that glucose phenotype is secondary to altered lipid metabolism in Sam68 ko mice.

Response

We thank the reviewer for the insightful comment and have performed additional experiments measuring serum triglyceride and free fatty acid levels in the feeding and 16 h-fasting mice; their levels were similar in Sam68^{f/f} and Sam68^{LKO} mice (Supplemental Figure 1i and 1j of our revised manuscript).

Minor concerns:

1. Figure 2. The glucagon dose for treating mouse hepatocyte was 100nM, which is too high for in vitro study. The lower dose of glucagon should be tested.

Response

We acknowledge the reviewer's concern. However, this glucagon concentration (100nM) has been widely used for *in vitro* studies²²⁻²⁷.

2. Poor quality of some immunoblotting with over-exposed images, including Figure 3a, 4a, 4h, 6a, 6b, 6e and 6c.

Response

We have now provided better quality images in Figure 3a, 6e, and 6c, and new data in Figures 4a, 4i (original 4h), 6a, and 6b.

3. Figure 4, author should test over-expression of Sam68 on CRTC2 protein half-life.

Response

Following the suggestion, we have performed additional experiment measuring the CRTC2 half-life in hepatocytes after Ad-GFP or Ad-Sam68 transduction; as expected, CRTC2 protein levels declined more slowly in Ad-Sam68-infected cells than in Ad-GFP-infected cells (Supplemental Figure 4e of our revised manuscript), which further supports that Sam68 promotes CRTC2 protein stability.

4. Figure 6, how about body weight after administrating shSam68 in ob/ob mice? Also lipid profile in these cohorts?

Response

We have addressed these two insightful questions by performing additional experiments. db/db mice were treated with control shRNA (sh-Scr) and Sam68 shRNA (sh-Sam68) and 3 weeks later, the body weights and serum triglyceride levels were measured. The body weights between the two treatment groups were similar (below, Figure 1A). The serum triglyceride level appeared slight lower in sh-Sam68 treated mice, however did not reach a significant difference (below,

Figure 1B).

5. How is Sam68 level induced in diabetic conditions? This is should be discussed or at least tested using in vitro experiments.

Response

We thank the reviewer for the insightful question. Please refer to our response to Reviewer #1/Comment [n].

References

- 1 Huot, M. E. *et al.* The Sam68 STAR RNA-binding protein regulates mTOR alternative splicing during adipogenesis. *Mol Cell* **46**, 187-199, doi:10.1016/j.molcel.2012.02.007 (2012).
- 2 Zhou, J. *et al.* Inhibition of Sam68 triggers adipose tissue browning. *J Endocrinol* **225**, 181-189, doi:10.1530/JOE-14-0727 (2015).
- 3 Hogan, M. F. *et al.* Hepatic Insulin Resistance Following Chronic Activation of the CREB Coactivator CRTC2. *J Biol Chem* **290**, 25997-26006, doi:10.1074/jbc.M115.679266 (2015).
- 4 Dentin, R. *et al.* Insulin modulates gluconeogenesis by inhibition of the coactivator TORC2. *Nature* **449**, 366-369, doi:10.1038/nature06128 (2007).
- 5 Wang, Y. *et al.* Targeted disruption of the CREB coactivator Crtc2 increases insulin sensitivity. *Proc Natl Acad Sci U S A* **107**, 3087-3092, doi:10.1073/pnas.0914897107 (2010).
- 6 Quintana-Portillo, R., Canfran-Duque, A., Issad, T., Sanchez-Margalet, V. & Gonzalez-Yanes, C. Sam68 interacts with IRS1. *Biochem Pharmacol* **83**, 78-87, doi:10.1016/j.bcp.2011.09.030 (2012).
- 7 Sanchez-Margalet, V., Gonzalez-Yanes, C., Najib, S., Fernandez-Santos, J. M. & Martin-Lacave, I. The expression of Sam68, a protein involved in insulin signal transduction, is enhanced by insulin stimulation. *Cell Mol Life Sci* **60**, 751-758 (2003).
- 8 Fumagalli, S., Totty, N. F., Hsuan, J. J. & Courtneidge, S. A. A target for Src in mitosis. *Nature* **368**, 871-874, doi:10.1038/368871a0 (1994).
- 9 Matter, N., Herrlich, P. & Konig, H. Signal-dependent regulation of splicing via phosphorylation of Sam68. *Nature* **420**, 691-695, doi:10.1038/nature01153 (2002).

- 10 Babic, I., Jakymiw, A. & Fujita, D. J. The RNA binding protein Sam68 is acetylated in tumor cell lines, and its acetylation correlates with enhanced RNA binding activity. *Oncogene* **23**, 3781-3789, doi:10.1038/sj.onc.1207484 (2004).
- 11 Cote, J., Boisvert, F. M., Boulanger, M. C., Bedford, M. T. & Richard, S. Sam68 RNA binding protein is an in vivo substrate for protein arginine N-methyltransferase 1. *Mol Biol Cell* **14**, 274-287, doi:10.1091/mbc.e02-08-0484 (2003).
- 12 Babic, I., Cherry, E. & Fujita, D. J. SUMO modification of Sam68 enhances its ability to repress cyclin D1 expression and inhibits its ability to induce apoptosis. *Oncogene* **25**, 4955-4964, doi:10.1038/sj.onc.1209504 (2006).
- 13 Bielli, P., Busa, R., Paronetto, M. P. & Sette, C. The RNA-binding protein Sam68 is a multifunctional player in human cancer. *Endocr Relat Cancer* **18**, R91-R102, doi:10.1530/ERC-11-0041 (2011).
- 14 Iijima, T. *et al.* SAM68 regulates neuronal activity-dependent alternative splicing of neurexin-1. *Cell* **147**, 1601-1614, doi:10.1016/j.cell.2011.11.028 (2011).
- 15 Huot, M. E., Brown, C. M., Lamarche-Vane, N. & Richard, S. An adaptor role for cytoplasmic Sam68 in modulating Src activity during cell polarization. *Mol Cell Biol* **29**, 1933-1943, doi:10.1128/MCB.01707-08 (2009).
- 16 Ramakrishnan, P. & Baltimore, D. Sam68 is required for both NF-kappaB activation and apoptosis signaling by the TNF receptor. *Mol Cell* **43**, 167-179, doi:10.1016/j.molcel.2011.05.007 (2011).
- 17 Fu, K. *et al.* Sam68 modulates the promoter specificity of NF-kappaB and mediates expression of CD25 in activated T cells. *Nat Commun* **4**, 1909, doi:10.1038/ncomms2916 (2013).
- 18 Zhou, J. *et al.* Regulation of vascular contractility and blood pressure by the E2F2 transcription factor. *Circulation* **120**, 1213-1221, doi:10.1161/CIRCULATIONAHA.109.859207 (2009).
- 19 Bielli, P. *et al.* The transcription factor FBI-1 inhibits SAM68-mediated BCL-X alternative splicing and apoptosis. *EMBO Rep* **15**, 419-427, doi:10.1002/embr.201338241 (2014).
- 20 Sellier, C. *et al.* Sam68 sequestration and partial loss of function are associated with splicing alterations in FXTAS patients. *EMBO J* **29**, 1248-1261, doi:10.1038/emboj.2010.21 (2010).
- 21 Lukong, K. E. & Richard, S. Sam68, the KH domain-containing superSTAR. *Biochim Biophys Acta* **1653**, 73-86, doi:10.1016/j.bbcan.2003.09.001 (2003).
- 22 Wang, Y. *et al.* Inositol-1,4,5-trisphosphate receptor regulates hepatic gluconeogenesis in fasting and diabetes. *Nature* **485**, 128-132, doi:10.1038/nature10988 (2012).
- 23 Koo, S. H. *et al.* The CREB coactivator TORC2 is a key regulator of fasting glucose metabolism. *Nature* **437**, 1109-1111, doi:10.1038/nature03967 (2005).
- 24 Zhang, W. S. *et al.* Inactivation of NF-kappaB2 (p52) restrains hepatic glucagon response via preserving PDE4B induction. *Nature communications* **10**, 4303, doi:10.1038/s41467-019-12351-x (2019).
- 25 Cao, W. *et al.* p38 Mitogen-activated protein kinase plays a stimulatory role in hepatic gluconeogenesis. *The Journal of biological chemistry* **280**, 42731-42737, doi:10.1074/jbc.M506223200 (2005).
- 26 Yang, W. *et al.* Glucagon regulates hepatic mitochondrial function and biogenesis through FOXO1. *J Endocrinol* **241**, 265-278, doi:10.1530/JOE-19-0081 (2019).

- 27 Mao, T. *et al.* PKA phosphorylation couples hepatic inositol-requiring enzyme 1alpha to glucagon signaling in glucose metabolism. *Proceedings of the National Academy of Sciences of the United States of America* **108**, 15852-15857, doi:10.1073/pnas.1107394108 (2011).

Reviewers' Comments:

Reviewer #1:

Remarks to the Author:

The authors have now provided new experimental evidence to address the issues raised to the original manuscript. In some cases, rational explanations of why they could not perform the proposed experiments at this time were given.

Reviewer #2:

Remarks to the Author:

The authors have addressed all my questions.

Reviewer #4:

Remarks to the Author:

This reviewer was asked to evaluate primarily the responses to Reviewer #3's prior comments, especially on the altered protein stability of CTRC2 by the manipulation of Sam68.

By in large, major questions raised previously by Reviewer #3 have been addressed quite convincingly by inclusion of new data and more targeted elaboration. In this reviewer's opinion, the data presented compellingly support authors' contention that the C-terminal of Sam68 via P5 interacts with CTRC2, which in turn prevents the ubiquitin ligase COP1 from binding and ubiquitinating CTRC2, thereby inhibiting proteasomal degradation of CTRC2; hence, loss of Sam68 destabilizes CTRC2 in a proteasome dependent manner. Meanwhile, this reviewer believes addressing the following minor issues should further strengthen the manuscript/study:

1. Figure 4a seems to show one genotype, contradicting to the legend claiming that "WT and Sam68(-/-) mice were treated with ...".
2. The CTRC2 half-lives derived from repeats of the cycloheximide chase experiment shown by Figure 4f could have been statistically compared between the two groups.

Response to Reviewers (R2)

Reviewer #4:

1. Figure 4a seems to show one genotype, contradicting to the legend claiming that “WT and Sam68(-/-) mice were treated with ...”.

Response:

This was an writing error. We thank the reviewer for pointing out. We have removed “and Sam68(-/-)”. The same error in the Supplementary Figure 4a legend has also been corrected.

2. The CTRC2 half-lives derived from repeats of the cycloheximide chase experiment shown by Figure 4f could have been statistically compared between the two groups.

Response:

We thank the reviewer for the important suggestion. Accordingly, we have quantified and statistically compared the half-lives ($t_{1/2}$) of CTRC2 in Sam68^{-/-} and WT primary hepatocytes; the $t_{1/2}$ in Sam68^{-/-} cells is >50% shorter than in WT cells ($p < 0.01$; $n = 3$ biologically independent cell samples). The figure 4f has been revised accordingly.